# Chemokine expression profile of an innate granuloma

**Megan E Amason**[1,2,3,4,5], **Cole J Beatty**[1,6], **Carissa K Harvest**[1,2,3,4,5], **Daniel R Saban**[1,6], **Edward A Miao**[1,2,4,5]*

[1]Department of Integrative Immunobiology, Duke University School of Medicine, Durham, United States; [2]Department of Ophthalmology, Duke University School of Medicine, Durham, United States; [3]Department of Molecular Genetics and Microbiology, Duke University School of Medicine, Durham, United States; [4]Department of Microbiology and Immunology, University of North Carolina at Chapel Hill, Chapel Hill, United States; [5]Department of Pathology, Duke University School of Medicine, Durham, United States; [6]Department of Cell Biology, Duke University School of Medicine, Durham, United States

## eLife assessment

This **valuable** study advances the understanding of granuloma formation by identifying a key chemokine receptors in containing infection by a specific species of bacteria. The evidence supporting this is **solid**, providing a spatial transcriptomic dataset spanning granuloma formation and resolution by a specific species of bacteria. The work should be of interest to microbiologists and immunologists.

*For correspondence:
edward.miao@duke.edu

**Abstract** Granulomas are defined by the presence of organized layers of immune cells that include macrophages. Granulomas are often characterized as a way for the immune system to contain an infection and prevent its dissemination. We recently established a mouse infection model where *Chromobacterium violaceum* induces the innate immune system to form granulomas in the liver. This response successfully eradicates the bacteria and returns the liver to homeostasis. Here, we sought to characterize the chemokines involved in directing immune cells to form the distinct layers of a granuloma. We use spatial transcriptomics to investigate the spatial and temporal expression of all CC and CXC chemokines and their receptors within this granuloma response. The expression profiles change dynamically over space and time as the granuloma matures and then resolves. To investigate the importance of monocyte-derived macrophages in this immune response, we studied the role of CCR2 during *C. violaceum* infection. *Ccr2⁻/⁻* mice had negligible numbers of macrophages, but large numbers of neutrophils, in the *C. violaceum*-infected lesions. In addition, lesions had abnormal architecture resulting in loss of bacterial containment. Without CCR2, bacteria disseminated and the mice succumbed to the infection. This indicates that macrophages are critical to form a successful innate granuloma in response to *C. violaceum*.

## Introduction

Granulomas are organized aggregates of immune cells defined by the presence of macrophages, with a variety of other features (i.e. necrosis and fibrosis) being more variable (**Warren, 1976**). The evolved function of the granuloma response is thought to be a protective mechanism by which immune cells sequester a foreign body or pathogen, walling-off the threat (**Pagán and Ramakrishnan, 2018**). Some pathogens are not successfully eliminated, however, leading to chronic granulomas that persist for

months or sometimes even years. New in vivo models are needed to study the complicated mechanisms that coordinate the formation of protective granulomas, in order to understand the events that lead to the successful clearance of pathogens that initiate this response.

We seek to identify environmental pathogens that have immense virulence capacity but are defeated by the innate immune system. *Chromobacterium violaceum* is one such pathogen that invades host cells and replicates in the intracellular niche, but only causes morbidity and mortality in immunocompromised animals or individuals (*Macher, 1982*). We discovered that during infection, wildtype (WT) C57BL/6 mice develop necrotic liver granulomas in response to this ubiquitous soil microbe (*Harvest et al., 2023*; *Maltez et al., 2015*). As soon as 1 day post-infection (1 DPI), liver microabscesses can be macroscopically visualized. These lesions are composed primarily of neutrophils until approximately 3–5 DPI, when, importantly, monocytes traffic into the area and form a mature granuloma starting at 5 DPI. Once the resulting macrophage zone surrounds the infected lesion, bacterial burdens begin to decrease, suggesting that granuloma macrophages are an important cell type for the clearance of *C. violaceum*. By 21 DPI, virtually all mice clear the infection and resolve the granuloma pathology, leaving small collagen scars in place of lesions (*Harvest et al., 2023*). Though we identified neutrophils and macrophages as the key immune players in this model, much remains to be learned about the cellular mechanisms that initiate formation of the granuloma in response to *C. violaceum*, and what signals instruct immune cells to organize within the granuloma architecture. Indeed, by studying the granuloma response that successfully clears *C. violaceum*, we hope to identify critical cellular mechanisms that underlie the basic biology of the granuloma response.

Within the granuloma response to *C. violaceum*, neutrophils and then macrophages migrate and assemble in an organized manner. Cellular movement, or chemotaxis, must be carefully regulated during tissue development, homeostasis, and inflammatory responses (*Hughes and Nibbs, 2018*). Chemotaxis is controlled by small, secreted proteins called chemokines that signal through transmembrane chemokine receptors. Since their discovery in the 1980s, approximately 50 chemokines are now appreciated for their role in cellular chemotaxis (*Zlotnik and Yoshie, 2012*). The temporal and spatial expression of chemokines and chemokine receptors dictate cellular trafficking, and dysregulation of these systems is linked to many diseases (*Turner et al., 2014*).

As more chemokines have been identified, there have been multiple revisions to their nomenclature, and now a systematic naming of chemokines and their receptors is in wide use. Chemokines have conserved cysteine residues, and the current naming system categorizes four subfamilies based on the arrangement of these N-terminal cysteines: CXC, CC, XC, and $CX_3C$ (*Zlotnik and Yoshie, 2000*). Though there are exceptions, most chemokines fit into one of these four subfamilies. Similarly, chemokine receptors fall into four subfamilies based on their chemokine ligand. The naming scheme has become complicated due to promiscuous ligand–receptor interactions, reassigning of mouse and human homologs after syntenic analysis, and divergent evolution of ligands in mice and humans (*Nomiyama et al., 2013*). Nonetheless, the detailed description of many chemokines and their receptors has been accomplished in both species. Herein, we focus on the mouse chemokines and their role in the *C. violaceum*-induced murine granuloma.

Inflammatory chemokines are those that are rapidly upregulated in the presence of infection or other inflammatory stimuli (*David and Kubes, 2019*). Several cell types can upregulate chemokines, creating a gradient of ligand that diffuses away from the point of infection. Still other cell types can respond to this gradient if/when they express the appropriate receptor. Furthermore, activated cells that migrate to the area can also upregulate expression of chemokines, creating a feed-forward loop to enhance cell recruitment. In addition to mediating chemotaxis, chemokines can induce a variety of other cellular responses including proliferation, oxidative burst, and even degranulation (*Hughes and Nibbs, 2018*). Lastly, it is now appreciated that chemokines also contribute to wound healing and resolution of inflammation, with coordinated efforts between neutrophils and macrophages to clean up debris and halt immune cell infiltration (*Soehnlein and Lindbom, 2010*).

Here, we use spatial transcriptomics to identify key genes that are upregulated in response to *C. violaceum*, and assess the importance of CCR2-dependent monocyte trafficking to the site of infection in the liver.

## Results

### Spatial transcriptomics of an innate granuloma

In our initial characterization of the granuloma response to *C. violaceum*, we used spatial transcriptomics (10x Genomics, Visium Platform) to identify genes that are upregulated at critical timepoints during infection, including 0.5, 1, 3, 5, 7, 10, 14, and 21 DPI (note: we excluded the 7 DPI timepoint from analysis because the granuloma in this capture area was not representative of typical 7 DPI granulomas histologically). A major advantage of this technology is the ability to conserve the spatial location of expression data by overlapping cDNA output with hematoxylin and eosin (H&E)-stained tissue sections (*Figure 1A*). Each capture area can collect nearly 5000 barcoded spots, each spot being 55 μm in diameter. Though this is not single-cell resolution, the dataset successfully identified 16 unique clusters with differentially expressed genes (*Figure 1B*), representing cell types (e.g. hepatocytes and endothelial cells), and also representing spatial elements (e.g. necrotic core center, etc.). We further characterized the clusters by assigning appropriate cell types based on each cluster's gene expression profile and its location within the granuloma (original characterization performed in *Harvest et al., 2023*, annotation shown in *Figure 1B–D*). Our previous analysis revealed that the clusters on the left of the UMAP (5: necrotic core center, 11: necrotic core-periphery, 9: coagulative necrosis, 0: macrophage, 8: coagulative necrosis-macrophage1, 6: coagulative necrosis-macrophage2, and 15: outside granuloma) all expressed varying levels of CD45 (*Harvest et al., 2023*). In contrast, the clusters on the right of the UMAP lacked CD45 but expressed higher levels of albumin. Though these hepatocyte clusters were abundantly present at each timepoint (not shown), the CD45-positive clusters were present to varying degrees. 10 DPI was the most enriched timepoint with all seven non-hepatocyte clusters present (*Figure 1C*). The sequencing depth varied between clusters, with areas of necrosis displaying relatively lower counts (*Figure 1—figure supplement 1A*). Cluster 0, which we previously identified as a macrophage-rich cluster, also had relatively lower counts (*Figure 1—figure supplement 1A, B*). Nevertheless, sufficient reads were obtained to reveal upregulated genes in these clusters, and the sctransform method was used to normalize the data such that biological heterogeneity was highlighted while minimizing technical variation associated with low counts (*Hafemeister and Satija, 2019*).

The spatial transcriptomics dataset was rich with candidate genes that could be critical for the successful granuloma response. Specifically, we were interested in the expression of chemokines and chemokine receptors that could be involved in the recruitment of key cell types, namely neutrophils and monocytes, to the site of infection within the liver. Indeed, immune cell trafficking is required for granuloma formation in various infectious and non-infectious models, and chemokines are the obvious candidates for facilitating this chemotaxis (*Chensue, 2013*).

To investigate various chemokines (*Table 1*) and chemokine receptors (*Table 2*), we used the Seurat package in RStudio to analyze gene expression over time and space. We used the SpatialFeaturePlot to assess relative gene expression within the granuloma at each timepoint (*Source code 1*). For example, *Pf4* (the murine homolog of *CXCL4*) is highly expressed at 10 DPI, corresponding with clusters 0, 6, 9, 11, and 15 (*Figure 1E*). Though chemokines and chemokine receptors are key facilitators of chemotaxis, other pro-inflammatory molecules such as damage-associated molecular patterns (DAMPs) and pathogen-associated molecular patterns (PAMPs) also direct cells to sites of inflammation. In fact, neutrophils respond to chemotactic molecules in a hierarchical manner, integrating a variety of signals and prioritizing end-target molecules (*David and Kubes, 2019*; *Kolaczkowska and Kubes, 2013*). Further demonstrating the complexity of chemotaxis, various adhesion molecules are also required for transmigration of cells out of the blood and into tissues. Indeed, we saw significant upregulation of a number of these genes in this model (*Table 3*), with many chemokines, chemokine receptors, and adhesion molecules appearing in the top twenty upregulated genes in several clusters (*Table 4*). Though these chemoattractive and adhesion molecules are likely involved and could be explored in future studies, in this paper we focus on the chemokines and their receptors.

### Expression of neutrophil-attractive chemokines

We observed high expression levels of chemokines involved in neutrophil trafficking (e.g. *Cxcl1* and *Cxcl2*) as early as 12 hr post-infection (0.5 DPI) (*Figure 2A, B*), which correlates with our previous data that neutrophils are the first immune cells to arrive in response to *C. violaceum* (*Harvest et al., 2023*). Two other ligands that also bind to CXCR2 are CXCL3 and CXCL5. In contrast to *Cxcl1* and

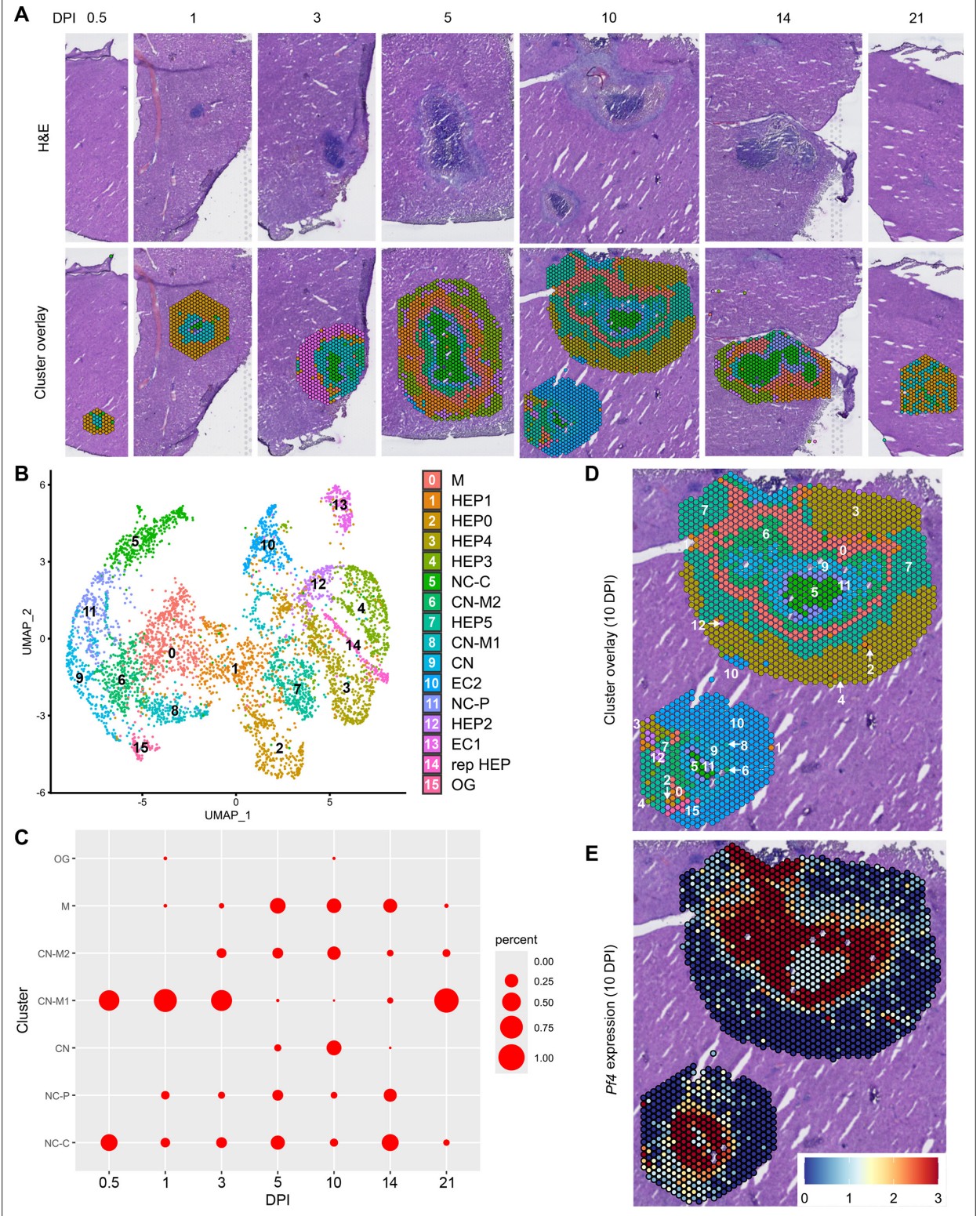

**Figure 1.** Spatial transcriptomics dataset reveals 16 unique clusters during infection with *C. violaceum*. (**A**) SpatialDimPlots showing hematoxylin and eosin (H&E) and cluster overlay of spatial transcriptomics data corresponding to various days post-infection (DPI). Each circle is an individual barcoded spot that is 55 µm in diameter. (**B**) UMAP plot of 16 unique clusters identified based on differentially expressed genes during the course of infection. Characterization of predominant cell types and/or location of each cluster (initial characterization performed in *Harvest et al., 2023*); macrophage zone (M), hepatocyte (HEP), representative HEP (rep HEP), necrotic core center (NC-C), NC-periphery (NC-P), coagulative necrosis (CN),

*Figure 1 continued on next page*

*Figure 1 continued*

CN-macrophage (CN-M), endothelial cell (EC), outside granuloma (OG). (**C**) Temporal prevalence of CD45⁺ clusters, calculated as proportion of spots represented by each cluster within each timepoint. (**D**) SpatialDimPlot at 10 DPI as in (**A**), showing cluster overlay and annotated with cluster identity. (**E**) SpatialFeaturePlot at 10 DPI, showing log-normalized expression of *Pf4* (murine homolog of *CXCL4*). ***Source code 1***. Streamlined code for analysis using RStudio.

The online version of this article includes the following figure supplement(s) for figure 1:

**Figure supplement 1.** Sequencing depth of samples and spatial expression of CXCR3 ligands.

*Cxcl2*, *Cxcl3*, and *Cxcl5* show delayed expression peaking around 10 DPI (***Figure 2C, D***). In addition to temporal differences, the spatial location of chemokine expression varies within the lesion. For example, at 5 DPI *Cxcl1* is expressed more toward the periphery of the lesion, while *Cxcl2* is expressed more toward the center (***Figure 2A, B***). For all of these ligands, expression is absent by 21 DPI, which correlates with the time at which the majority of mice clear the infection. Therefore, although these four chemokines all bind to CXCR2, they clearly demonstrate the complexity of different temporal and spatial expression profiles over the course of infection.

## Expression of monocyte-attractive chemokines

We also investigated chemokines and receptors involved in monocyte trafficking (e.g. *Ccl2*, *Ccl7*, and *Ccl12*). Though all three of these ligands bind to CCR2, they had vastly different expression levels through the course of infection (***Figure 3***). *Ccl2* was the most highly upregulated, while *Ccl12* was expressed only at low levels, and *Ccl7* expression was somewhere in between (***Figure 3A–C***). Similar to the chemokines involved in neutrophil trafficking, these ligands are not expressed by 21 DPI.

## Compilation of chemokine and receptor expression data

In order to summarize our findings in a way that facilitates comparisons, we used the SpatialFeaturePlot to visually rank the expression intensity of each chemokine and receptor as absent, low, medium, or high over the course of infection. Each rank was based on both the intensity of expression and the relative number of spots that expressed the gene. For example, *Cxcl1* expression was ranked as medium at 0.5 DPI, and ranked as high at 1 and 3 DPI based on the large presence of orange and red spots (***Figure 2A***). In contrast, *Cxcl3* was ranked as absent at 0.5 DPI, low at 1 DPI, and medium at 3 DPI based on the fewer spots that were orange or red (***Figure 2C***). We depicted these ranks as qualitative heatmaps (***Figure 4A–D***). The relative expression of various chemokines (***Figure 4—figure supplements 1–4***) was much greater than the relative expression of their receptors (***Figure 4—figure supplements 5 and 6***), which is expected because large quantities of chemokines are needed to create gradients in tissues, but comparatively low expression of chemokine receptors is sufficient to enable trafficking of cells that express the receptors. Therefore, we changed the scale to best visualize receptor expression.

One aspect of chemokine biology that makes understanding their function complicated is the promiscuity of certain ligands for multiple receptors, and vice versa. For example, CCL3, which is highly upregulated during infection with *C. violaceum*, can bind to CCR1 (along with several other chemokines), and CCL3 can also bind to CCR5 (again, along with several other chemokines). This promiscuity often makes it challenging to determine what unique or redundant roles each chemokine and chemokine receptor are playing. In order to simplify and graphically depict ligand and receptor interactions that seem relevant to the *C. violaceum*-induced granuloma, we listed the ligands that bind to the receptors that were expressed (***Figure 4C, D***). We colored each respective ligand based on its maximum expression ranking, regardless of the timepoint. This visualization allows for easier generation of hypotheses from this complex dataset.

## Weakly expressed chemokines suggest that certain immune cells are dispensable

The chemokines that are not present or are only weakly expressed can also be informative (***Figure 4A, B***). Two chemokines that are important for migration to the lung, *Cxcl15* and *Cxcl17*, are both absent (as expected). Still other chemokines that are important for migration to the skin, lymph nodes, and mucosal tissues are also absent, namely *Ccl17*, *Ccl27b*, *Ccl21b-c*, and *Ccl28*, respectively (also as

**Table 1.** Expression level of chemokine ligands during infection with *C. violaceum*.
Expression was visually ranked as absent, low, medium, or high based on SpatialFeaturePlots.
Maximum expression rank recorded here. Table generated from *David and Kubes, 2019*; *Hughes and Nibbs, 2018*; *Sokol and Luster, 2015*; *Zlotnik and Yoshie, 2000*; *Zlotnik and Yoshie, 2012*.
Lymph node (LN); natural killer cell (NK); NK T cell (NKT); innate lymphoid cell (ILC); dendritic cell (DC).

| Ligand | Max expression | Alias and main functions |
|---|---|---|
| *Cxcl1* | High | (NAP-3) Neutrophil migration |
| *Cxcl2* | High | (MIP-2) (MIP2-α) Neutrophil migration; 90% identical to *Cxcl1*; involved in wound healing |
| *Cxcl3* | High | (MIP2-β) Neutrophil migration; migration and adhesion of monocytes |
| *Cxcl4* | High | (*Pf4*) Neutrophil and monocyte migration; released by platelets; wound repair and coagulation; angiogenesis |
| *Cxcl5* | High | (LIX) Neutrophil migration; connective tissue remodeling |
| *Cxcl9* | High | Th1, CD8, NK, monocyte migration; closely related to CXCL10 and CXCL11 |
| *Cxcl10* | High | Th1, CD8, NK, monocyte migration |
| *Cxcl11* | Absent | Th1, CD8, NK, monocyte migration |
| *Cxcl12* | High | (SDF-1) Lymphocyte migration; bone marrow homing |
| *Cxcl13* | Low | B cell migration within follicles of lymphoid tissues; highly expressed in liver, spleen, LN |
| *Cxcl14* | Low | Monocyte migration to skin; potent activator of DC |
| *Cxcl15* | Absent | Neutrophil migration during inflammation of lungs |
| *Cxcl16* | Med | NKT and ILC migration and survival; found in red pulp of the spleen |
| *Cxcl17* | Absent | Monocyte and DC migration in the lung |
| *Ccl1* | Absent | (TCA3) T cell trafficking |
| *Ccl2* | High | (MCP1) Monocyte trafficking |
| *Ccl3* | High | (MIP-1α) Macrophage and NK cell migration |
| *Ccl4* | High | (MIP-1β) Macrophage and NK cell migration |
| *Ccl5* | High | (RANTES) Macrophage and NK cell migration; also chemotactic for T cells, eosinophils, basophils |
| *Ccl6* | High | (C10) Myeloid cell differentiation; monocyte, T cell, and eosinophil chemotaxis |
| *Ccl7* | Med | (MCP3) (MARC) Monocyte mobilization |
| *Ccl8* | Med | (MCP2) Th2 response; skin homing |
| *Ccl9* | High | (MIP-1γ) (MRP-2) DC migration |
| *Ccl11* | Low | (Eotaxin) Eosinophil and basophil migration; selectively recruits eosinophils |
| *Ccl12* | Low | (MCP5) Inflammatory monocyte trafficking |
| *Ccl17* | Absent | (ABCD2) (TARC) T cell chemotaxis; lung and skin homing |

*Table 1 continued on next page*

Table 1 continued

| Ligand | Max expression | Alias and main functions |
| --- | --- | --- |
| Ccl19 | Med | (MIP-3β) T cell and DC migration to LN |
| Ccl20 | Low | (MIP-3α) Th17 responses; B cell and DC homing to gut-associated lymphoid tissue |
| Ccl21a | Med | (TCA4) T cell and DC migration to LN |
| Ccl21b | Absent | Very similar to Ccl21a |
| Ccl21c | Absent | Identical to Ccl21b |
| Ccl22 | Low | (ABCD1) Th2 response and migration; monocyte, DC, NK migration; produced by monocytes and DC |
| Ccl24 | Med | (MPIF-2) (Eotaxin-2) Eosinophil and basophil migration |
| Ccl25 | Low | (TECK) T cell homing to gut; T cell development; thymocyte, macrophage, and DC migration |
| Ccl26 | Absent | (Eotaxin-3) Eosinophil and basophil migration |
| Ccl27a | Low | T cell migration to skin |
| Ccl27b | Absent | T cell migration to skin |
| Ccl28 | Absent | (MEC) T and B cell migration to mucosal tissues |
| Cx3cl1 | Low | (Fractalkine) NK, monocyte, and T cell migration |
| Xcl1 | Low | (Lymphotactin) Cross-presentation by CD8[+] DCs |

expected). Such negative data provide stronger confidence in the positive expression data for other chemokines.

In our previous studies, we showed that the adaptive immune response is dispensable to successfully form granulomas around, and then to eradicate, *C. violaceum* (*Harvest et al., 2023*). In agreement with those findings, several chemokines involved in T cell trafficking are absent or only expressed at low levels (i.e. *Cxcl11*, *Ccl1*, *Ccl22*, and *Ccl25*) (*Figure 4A, B*). On the other hand, other chemokines involved in T cell trafficking such as *Cxcl9* and *Cxcl10* are highly expressed during the first few days of infection, as is their receptor *Cxcr3* (*Figure 4A, C*). During primary infection, T cell recruitment is not essential for clearance and we found that T cells are not recruited in large numbers (*Harvest et al., 2023*). However, *Cxcl9* and *Cxcl10* could play a more important role during a secondary infection that involves the adaptive immune response. It is a curious observation that T cells are dispensable during primary infection because in *Mycobacterium tuberculosis*-induced granulomas, CD4[+] T helper type 1 (Th1) cells are required to stimulate the antibacterial activity of macrophages (*Pagán and Ramakrishnan, 2018*). A key difference between granuloma formation in response to *C. violaceum* compared to *M. tuberculosis* could be that *M. tuberculosis* is able to intracellularly infect macrophages, whereas *C. violaceum* is unable to circumvent pyroptosis of macrophages.

We did not observe basophils or eosinophils histologically during infection with *C. violaceum*, and this was again supported by the absence or low expression of chemokines involved in trafficking of these cell types (i.e. *Ccl11*, *Ccl24*, and *Ccl26*) (*Figure 4B*). CCR3, which is expressed mainly by eosinophils, plays a major role in the granuloma response to parasitic *Schistosoma mansoni* eggs (*Chensue, 2013*). During infection with *C. violaceum*, however, *Ccr3* is not expressed at any timepoint (*Figure 4D*), further supporting that eosinophils are not involved in the granuloma response to *C. violaceum*. Furthermore, granulomas that form in response to *M. tuberculosis* often contain follicular dendritic cells which secrete CXCL13 to recruit B cells via CXCR5 (*Domingo-Gonzalez et al., 2016*). However, *Cxcl13* is expressed at low levels, and *Cxcr5* is absent in the *C. violaceum* model (*Figure 4A, C*). These examples reveal chemokines that are likely dispensable in the context of *C. violaceum*.

**Table 2.** Expression level of chemokine receptors during infection with *C. violaceum*.
Expression was visually ranked as absent, low, medium, or high based on SpatialFeaturePlots.
Maximum expression rank recorded here. Table generated from *David and Kubes, 2019*; *Hughes and Nibbs, 2018*; *Sokol and Luster, 2015*; *Zlotnik and Yoshie, 2000*, *Zlotnik and Yoshie, 2012*.
Natural killer cell (NK); innate lymphoid cell (ILC); dendritic cell (DC); plasmacytoid DC (pDC); lymph node (LN); red blood cell (RBC).

| Receptor | Max expression | Alias, cellular expression, and main functions |
|---|---|---|
| *Cxcr1* | Absent | (IL8R-α) Neutrophil, monocyte, NKs, mast cell, basophil, CD8 T cells; neutrophil migration and activation |
| *Cxcr2* | Med | (IL8R-β) Neutrophil, monocyte, NKs, mast cell, basophil, CD8 T cells; B cell and neutrophil migration; neutrophil egress from BM |
| *Cxcr3* | Med | Various T cells, NKs, pDCs, B cells; effector T cell migration and activation |
| *Cxcr4* | Med | Most leukocytes; bone marrow homing and retention |
| *Cxcr5* | Absent | B cells, T cells; T and B cell migration within LN to B cell zones |
| *Cxcr6* | Med | Various T cells, ILCs, NKs, plasma cells; T cell and ILC function |
| *Ccr1* | High | Monocyte, macrophage, neutrophil, Th1, basophil, DC |
| *Ccr2* | High | Monocyte, macrophage, Th1, DC, basophil, NK; monocyte migration, Th1 immunity |
| *Ccr3* | Absent | Highly expressed on eosinophils and basophils; allergic airway; eosinophil trafficking |
| *Ccr4* | Absent | Various T cells, monocytes, B cells, DCs; T cell homing to skin and lung |
| *Ccr5* | High | Monocytes, macrophages, various T cells, NK, DC, neutrophils, eosinophils; adaptive immunity |
| *Ccr6* | Absent | Various T cells, DCs, NKs; DC and B cell maturation and migration; adaptive immunity |
| *Ccr7* | Med | Various T cells, DCs, B cells; migration of adaptive lymphocytes and DCs to lymphoid tissues |
| *Ccr8* | Absent | Various T cells, monocytes, macrophages; surveillance in skin; expressed in the thymus |
| *Ccr9* | Absent | T cells, thymocytes, B cells, DCs, pDCs; T cell migration to gut; key regulator of thymocyte migration and maturation |
| *Ccr10* | Absent | T cells, melanocytes, plasma cells; immunity at mucosal sites, especially skin |
| *Xcr1* | Low | DCs; antigen cross-presentation |
| *Cx3cr1* | Low | Monocytes, macrophages, microglia, DCs, T cells; migration and adhesion of leukocytes; marker of anti-inflammatory monocytes; thought to promote a patrolling phenotype and pro-survival signals |

Atypical receptors

*Table 2 continued on next page*

Table 2 continued

| Receptor | Max expression | Alias, cellular expression, and main functions |
| --- | --- | --- |
| Ackr1 | Low | (DARC) RBCs, endothelial cells, neurons; chemokine scavenging, neutrophil transmigration; chemokine transcytosis on lymphatic endothelium and RBCs |
| Ackr2 | Low | Endothelial cells, DCs, B cells, macrophages; chemokine scavenging |
| Ackr3 | Low | (Cxcr7) Stromal cells, B cells, T cells, neurons, mesenchymal cells; pro-survival, adhesion, shaping CXCR4 gradients; involved in CXCR4 gradients |
| Ackr4 | Low | (Ccrl1) Epithelial cells, leukocytes, astrocytes, microglia; chemokine scavenging and transcytosis; chemokine scavenging in thymus |
| Ccrl2 | High | Chemokine receptor-like protein; binds chemerin; related to CCR1; expressed on neutrophils and monocytes |

## Comparison of neutrophil- and monocyte-recruiting chemokines

To compare chemokines involved in neutrophil recruitment or monocyte recruitment, we further characterized *Cxcl1* and *Ccl2*, respectively (*Figure 5*). When comparing their SpatialFeaturePlots, *Cxcl1* and *Ccl2* had unique expression profiles corresponding to different cluster identities (*Figures 2A and 3A*). To more easily visualize these differences in expression, we generated UMAP plots and violin plots (*Figure 5A–D*). Though there is some overlap, suggesting that some clusters express both *Cxcl1* and *Ccl2*, there are also some clusters that appear to express only one or the other (*Figure 5A, B*). For example, cluster 14 (a cluster enriched for hepatocytes) expressed high levels of *Cxcl1* but only low levels of *Ccl2* (*Figure 5C, D*). Furthermore, there are interesting differences in temporal expression; *Cxcl1* is highly expressed at 1 DPI while *Ccl2* expression peaks at 3 DPI (*Figure 5E, F*). Though gene expression does not necessarily correlate with the timing and intensity of protein expression, we expect CXCL1 and CCL2 protein levels to accumulate over time, which would allow proper chemokine gradients to form. Altogether, these data corroborate our previous findings that neutrophils traffic to the liver within 1 DPI, and monocytes traffic and form granulomas beginning at 3 DPI.

## Neutrophil chemotaxis

We next wanted to investigate whether the upregulated neutrophil-recruiting chemokines are important during infection. However, there are many challenges when studying chemokines. As previously mentioned, ligands and receptors often show promiscuity in that one receptor may bind multiple ligands, which makes it difficult to completely abrogate chemotaxis through inhibiting a single ligand. Furthermore, although chemokine-specific antibodies exist (*Fox et al., 2009*; *Mollica Poeta et al., 2019*; *Vales et al., 2023*), neutralizing such large quantities of ligand can be challenging. Therefore, instead of attempting to block chemokine ligands, we chose to target chemokine receptors. In fact, the promiscuity of ligands and receptors means that targeting one chemokine receptor has the potential to impact more than one ligand of interest (*Figure 4C, D*). Nevertheless, targeting receptors is also challenging due to poor solubility of many receptor antagonists (*Li et al., 2019*).

During infection with *C. violaceum*, neutrophils appear in the liver within 1 DPI. However, it is still unclear what signals initiate their migration into the liver. Though a large number of neutrophils are already present in the blood during homeostasis, additional neutrophils expressing CXCR2 exit the bone marrow in response to endothelial cell-derived CXCL1 and CXCL2 (*David and Kubes, 2019*). Furthermore, tissue-resident macrophages can also express CXCL1, CXCL2, and various leukotrienes in response to infection (*Soehnlein and Lindbom, 2010*). Though *Cxcr2* knockout mice exist, they have abnormalities (*Cacalano et al., 1994*). Therefore, to assess the role of CXCL1, CXCL2, CXCL3, and CXCL5 in neutrophil trafficking during infection with *C. violaceum*, we used a CXCR2 inhibitor. Reparixin is an allosteric inhibitor of CXCR1 and CXCR2 that has been shown to inhibit neutrophil

**Table 3.** Expression level of selected proteins and receptors during infection with *C. violaceum*. Expression was visually ranked as absent, low, medium, or high based on SpatialFeaturePlots. Maximum expression rank recorded here. Table generated from *Bui et al., 2020*; *David and Kubes, 2019*; *Parks et al., 2004*; *Wang et al., 2018*. Dendritic cell (DC); plasmacytoid DC (pDC); Kupffer cell (KC); natural killer cell (NK); syndecan 1 (SDC1).

| Other | Max expression | Alias, cellular expression, and main functions |
|---|---|---|
| *Fpr1* | High | (Formyl peptide receptor 1) Expressed on myeloid cells and lymphocytes; widely expressed by neutrophils, eosinophils, basophils, monocytes, and platelets (among others); involved in leukocyte chemotaxis and activation |
| *Fpr2* | Med | (Formyl peptide receptor 2) Expressed on neutrophils, eosinophils, monocytes, macrophages, T cells; involved in leukocyte chemotaxis and activation |
| *C5ar1* | Med | (Complement C5a receptor 1) Expressed on basophils, DCs, mast cells, non-immune cells; involved in leukocyte chemotaxis and activation |
| *Ltb4r1* | Low | (Leukotriene B4 receptor) Expressed on neutrophils, macrophages, T cells; involved in leukocyte chemotaxis and activation |
| *Cmklr1* | Low | (Chemerin chemokine-like receptor 1) Expressed mainly on myeloid cells; present in thymus, bone marrow, spleen, fetal liver, and lymphoid organs; involved in migration of macrophages, DCs, and pDCs |
| *Mmp2* | High | (Gelatinase A) Inactivates CXCL12, CCL7; degrades S100A9 |
| *Mmp8* | Med | (Neutrophil collagenase) Stored in secondary granules; cleaves and enhances CXCL5; inactivates CXCL-9 and CXCL-10 |
| *Mmp9* | High | (Gelatinase B) Mainly expressed by neutrophils; cleaves and enhances CXCL5; cleaves SDC1 to promote neutrophil infiltration; inactivates CXCL4 and CXCL1; inactivates CXCL-9 and CXCL-10; upregulated during respiratory epithelial healing; also expressed by KCs |
| *Mmp12* | High | (Macrophage elastase) Activates TNF release from macrophages |
| *Mmp13* | Med | (Collagenase 3) Inactivates CXCL-12; inactivates CCL2, CCL8, CCL13 |
| *Itgam* | Med | (CR3A) (Cd11b) Regulates adhesion and migration of monocytes, granulocytes, macrophages, NKs; involved in complement system |
| *Mif* | High | (Macrophage migration inhibitory factor) Binds to CXCR2 and CXCR4 to promote chemotaxis of leukocytes |
| *Icam1* | High | (Intracellular adhesion molecule 1) Promotes leukocyte migration from circulation to sites of inflammation |
| *S100a8* | High | Heterodimerizes with S100a9; involved in leukocyte recruitment and inflammation |
| *S100a9* | High | Heterodimerizes with S100a8; involved in leukocyte recruitment and inflammation |

**Table 4.** Top 20 differentially expressed genes per cluster.

The FindAllMarkers function was used to identify the top differentially expressed genes for each cluster across all timepoints. Genes were sorted from highest to lowest average log2 fold change (avg_log2FC) values within each cluster. Genes of interest shown in red. Full dataset found in *Table 4—source data 1*.

| M | HEP1 | HEP0 | HEP4 | HEP3 | NC-C | CN-M2 | HEP5 | CN-M1 | CN | EC2 | NC-P | HEP2 | EC1 | rep HEP | OG |
|---|---|---|---|---|---|---|---|---|---|---|---|---|---|---|---|
| 0 | 1 | 2 | 3 | 4 | 5 | 6 | 7 | 8 | 9 | 10 | 11 | 12 | 13 | 14 | 15 |
| Mmp2 | Spink1 | Mup11 | Acot3 | Mup21 | Ewsr1 | Col11a1 | Gm31583 | Ptgs2 | F13a1 | Hbb-bt | Hcar2 | Elovl3 | Derl3 | Ly6d | Ccl8 |
| Aebp1 | Gstm3 | Mup17 | Cyp4a14 | Elovl3 | Parp10 | Ptprn | Mpo | Il11 | Cxcl3 | Hba-a1 | Cxcl3 | Cyp4a12b | 3930402G23Rik | Moxd1 | Gm32468 |
| Olfml3 | Ifi27l2b | Cyp2b13 | Cyp2c69 | Serpina1e | Fth1 | Ccl11 | Gdf10 | Cxcl10 | Pf4 | Hba-a2 | Ptges | Hsd3b5 | Hyou1 | BC049987 | Kdelr3 |
| Cd74 | Klk1b4 | Mup12 | Sult2a1 | Cib3 | Ptprc | Prnd | Cd207 | Cxcl9 | Mmp9 | Hbb-bs | Tnf | Gm32468 | Sult3a1 | Esco2 | Hbb-bt |
| Pacs2 | Vnn3 | Mup16 | Cyp2a4 | Sds | Csf3r | Cthrc1 | Gck | Il6 | Ptges | mt-Atp8 | Ccl4 | Lhpp | Sdf2l1 | Gsta1 | Cyp1b1 |
| Ngp | Cib3 | Mup7 | Cyp4a10 | Mfsd2a | Pacs2 | Gpnmb | Cyp8b1 | Serpine1 | Cstdc4 | mt-Nd4l | Cxcl2 | Cyp4a12a | Apcs | Cdkn3 | Lgals1 |
| Ewsr1 | Cdh1 | Mup1 | Sult2a2 | Acmsd | Lyn | Actg2 | Abcd2 | Hspa1a | Gpr84 | Malat1 | Il1f9 | Fitm1 | Pdia4 | Chrna4 | Vwf |
| Clu | Frzb | Mup3 | Fmo3 | Slc22a7 | Osbpl9 | Fbln2 | 1700001C19Rik | Adm | Itgam | mt-Nd3 | Fth1 | Oat | Dnajb9 | Nat8 | Cthrc1 |
| Cdk11b | Spon2 | Cyp2b9 | Slc16a5 | Hectd1 | Col12a1 | Sulf1 | Defb1 | Gm15056 | Fpr2 | mt-Nd5 | Ccl3 | Slc1a2 | A1bg | Nat8f5 | Cpe |
| Parp8 | Snta1 | Cyp7b1 | Cyp2b9 | Slc10a2 | Iqgap1 | Sulf1 | Prox1os | Nos2 | Adam8 | mt-Nd2 | Slfn4 | Cyp2a5 | Prg4 | Mup1 | Pcdh17 |
| Nisch | Wfdc2 | Mup20 | A1bg | Selenbp2 | Clk1 | Mmp13 | Socs2 | Gbp5 | Lyz2 | mt-Co2 | Asprv1 | Tuba8 | Gm26917 | Thrsp | Gm32468 |
| Cpxm1 | Gstm2 | Gm13775 | Cyp2c40 | Mmd2 | Lilr4b | Sfrp1 | Bik | Olr1 | Clec4d | Elane | Slc7a11 | Cyp2c55 | Mt2 | Gm32468 | Ccdc80 |
| Poglut1 | Spic | mt-Atp8 | Slc22a27 | G6pc | Thrap3 | Fkbp10 | Afmid | Rnd1 | Cav1 | Gm26917 | Acod1 | Rhbg | Cyp17a1 | Cdca3 | Mrc2 |
| Col6a2 | Tmem268 | Mup9 | Cyp2c37 | Arl4d | Stip1 | Lox | Rad51b | Retnlg | Mmp8 | mt-Atp6 | Slpi | Slc13a3 | Creld2 | Hebp2 | Hbb-bs |
| Loxl1 | Tstd1 | Serpina3m | Cyp2c38 | Kcnk5 | Fbxl5 | Acta2 | 1810059H22Rik | Il1a | Il1f9 | mt-Nd1 | Ccrl2 | Cyp7a1 | Vnn1 | Ect2 | Ccbe1 |
| Gpx3 | Prelp | Itih4 | Acot1 | Lpin1 | Zfp207 | Col15a1 | Tmem25 | F3 | Fpr1 | Gm26917 | Il1rn | Glul | Hist1h4h | Pbk | mt-Nd1 |
| Col1a1 | Slc39a4 | Slco1a1 | Etnppl | Tat | Klf2 | Nbl1 | Angptl6 | Cxcl2 | Capg | Gm29966 | Slc25a37 | Slc1a4 | Rcan2 | Cdc20 | Plxdc2 |
| Igha | Mki67 | Cyp2b10 | Gstt3 | Upp2 | Hck | Col5a2 | Fam89a | Procr | Stfa2l1 | mt-Co3 | Mmp12 | Rdh16 | Hspa5 | Gpam | Nat8f5 |
| Ikbkb | Cdk1 | Car3 | Gm13775 | Pck1 | Rhob | Col5a1 | Mug1 | AA467197 | Pqlc3 | Gm42418 | Clec4e | Serpina7 | mt-Atp6 | Nek2 | Chrna4 |
| Rpl4 | Mcm5 | Fbxo31 | Ptgds | Fam47e | Lilrb4a | Tnc | Ccl27a | Plaur | Pdpn | mt-Co1 | Il1b | Cyp1a2 | mt-Co2 | Aurka | Snhg18 |

The online version of this article includes the following source data for table 4:

**Source data 1.** Top differentially expressed genes for each cluster across all timepoints.

trafficking during ischemia–reperfusion injury and acid-induced acute lung injury (*Bertini et al., 2004*; *Zarbock et al., 2008*; *Hosoki and Sur, 2018*). We pre-treated mice with reparixin or saline (PBS) 1 day before infection, then infected mice with *C. violaceum* followed by daily treatment with reparixin or PBS (*Figure 5—figure supplement 1A*). We then harvested livers and spleens at 3 DPI to assess bacterial burdens. Though there was no difference in CFU for the liver, a few CFU were recovered from the spleens of two reparixin-treated mice (*Figure 5—figure supplement 1B*), which, though this was not statistically significant, is unusual for WT mice. Based on these results, we hypothesized that reparixin would have a stronger effect at 1 DPI (*Figure 5—figure supplement 1C*), before the infection causes excessive damage to the liver. At 1 DPI, we again saw no difference in bacterial burdens in the liver of reparixin-treated mice (*Figure 5—figure supplement 1D*). To verify that reparixin affected neutrophil numbers in the liver and spleen, we used flow cytometry to quantify Ly6G$^+$ neutrophils (*Figure 5—figure supplement 1E*). We observed differences in the number of neutrophils between PBS-treated female and male mice, so data were analyzed disaggregated for sex. Though reparixin might have caused a subtle decrease in neutrophil numbers in the liver and spleen at 1 DPI, the results were variable between mice (*Figure 5—figure supplement 1F, G*). In our hands, reparixin was poorly soluble in PBS, which could account for some of the variability. Because monocytes also express CXCR2, albeit to a much lesser extent than neutrophils, we also stained for CD68. There was no marked difference in macrophage numbers in the liver or spleen between PBS- and reparixin-treated mice (*Figure 5—figure supplement 1H, I*).

Altogether, it is clear that reparixin was not a successful inhibitor of neutrophil recruitment during infection with *C. violaceum*. The role of CXCR1/2 and their ligands could be further studied using

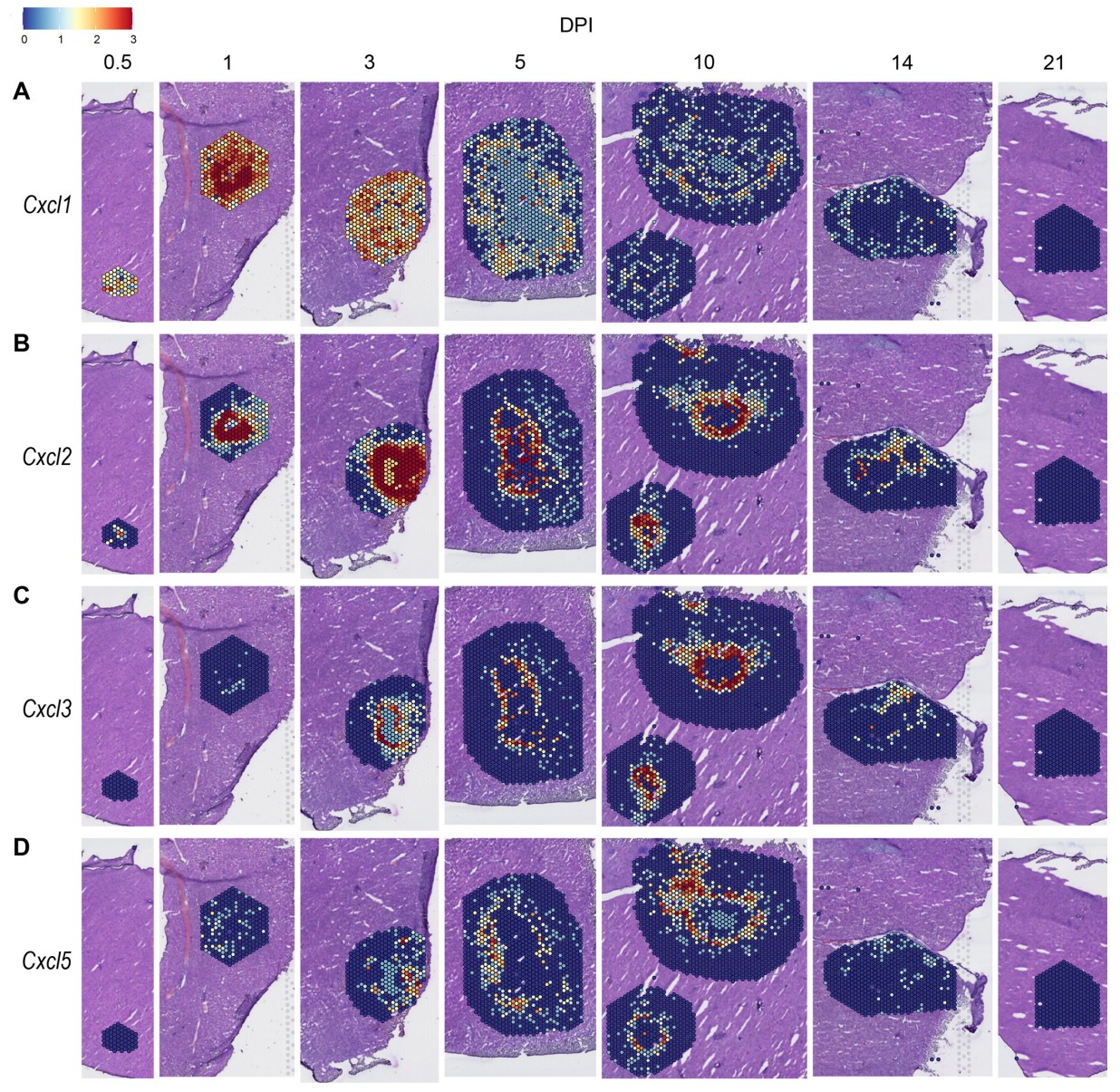

**Figure 2.** Chemokines involved in neutrophil recruitment are upregulated during infection. SpatialFeaturePlots displaying normalized gene expression data of CXCR2 ligands (i.e. *Cxcl1*, *Cxcl2*, *Cxcl3*, and *Cxcl5*) at various days post-infection (DPI). Scale set at 0–3.0 expression.

knockout mice. Regardless, other chemoattractants likely contribute to neutrophil recruitment as well. Indeed, neutrophils migrate in response to a variety of pro-inflammatory DAMPs and PAMPs (***Kolaczkowska and Kubes, 2013***). Importantly, formyl peptide receptors (FPRs) such as FPR2 promote neutrophil migration in response to bacterial infection in the liver (***Lee et al., 2023***). In support of this, FPRs are upregulated in this model (***Table 3***).

## CCR2 is essential for monocyte trafficking and defense against *C. violaceum*

Previously, we noticed that the appearance of organized macrophages at approximately 5 DPI correlates with a subsequent decrease in bacterial burdens (***Harvest et al., 2023***). We also observed that *Nos2*⁻/⁻ mice, which lack the ability to express inducible nitric oxide synthase (iNOS), succumb to infection beginning at 7 DPI, a timepoint when the granuloma matures with a thicker macrophage ring (***Harvest et al., 2023***). Though neutrophils can also express iNOS (***Saini and Singh, 2018***), these data

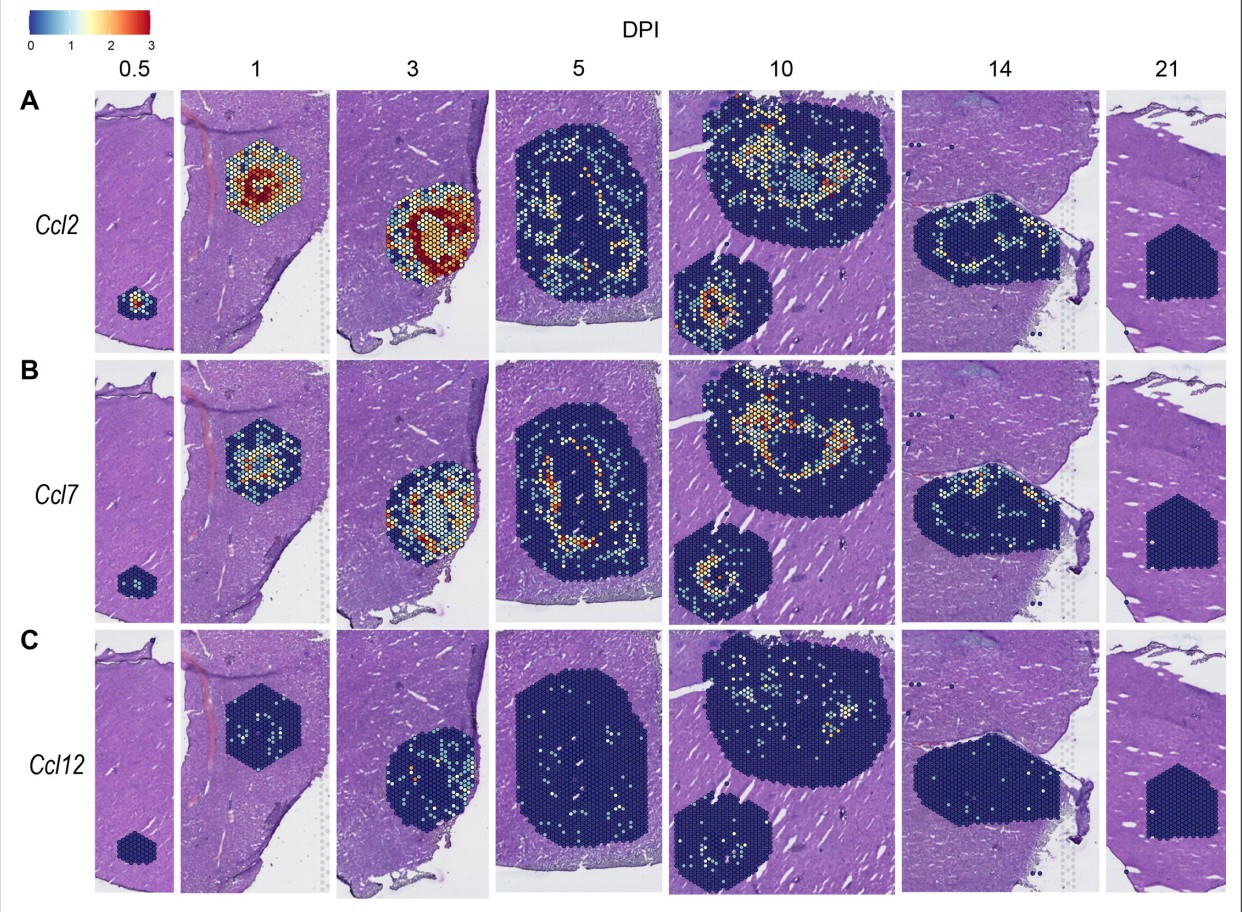

**Figure 3.** Chemokines involved in monocyte recruitment are upregulated during infection. SpatialFeaturePlots displaying normalized gene expression data of CCR2 ligands (i.e. *Ccl2*, *Ccl7*, and *Ccl12*) at various days post-infection (DPI). Scale set at 0–3.0 expression.

suggested that macrophages are playing a critical protective role. We therefore hypothesized that monocyte trafficking to the site of infection is a key event in clearing the infection. There are several candidate chemokines that could attract monocytes to the site of infection, and these chemokines bind to several different receptors (*Table 1*, *Figure 4D*). We chose to focus on the chemokine receptor CCR2 because of its known role in monocyte migration out of the bone marrow (*Serbina and Pamer, 2006*). Importantly, *Ccr2$^{-/-}$* mice have intact tissue-resident macrophage populations but are unable to recruit additional monocytes in the event of infection (*Kurihara et al., 1997*).

To assess the role of monocyte trafficking to lesions in the liver, we infected *Ccr2$^{-/-}$* mice with *C. violaceum*. Strikingly, *Ccr2$^{-/-}$* mice were highly susceptible and succumbed to infection beginning at 5 DPI, with all mice dying by 9 DPI (*Figure 6A*), which is more severe than the phenotype in *Nos2$^{-/-}$* mice (*Harvest et al., 2023*). This is in contrast to *Yersinia pseudotuberculosis* models in which deletion of *Ccr2* has the opposite phenotype, and loss of monocytes is actually protective (*Zhang et al., 2018*). This also contrasts with *M. tuberculosis* models where loss of *Ccr2* has no effect on survival in some contexts (*Domingo-Gonzalez et al., 2016*; *Scott and Flynn, 2002*). At 5 DPI, *Ccr2$^{-/-}$* mice had increased liver burdens (*Figure 6B*), and bacterial dissemination into the spleen (*Figure 6C*). We also observed that *Ccr2$^{-/-}$* mice had abnormal lesions which were more numerous and larger than the lesions of WT mice (*Figure 6D*).

We used flow cytometry to assess macrophage (CD68$^+$) and neutrophil (Ly6G$^+$) numbers in the liver, spleen, and blood of mice at 5 DPI (*Figure 6E*, *Figure 6—figure supplement 1*). Uninfected WT and uninfected *Ccr2$^{-/-}$* mice had a similar frequency of macrophages in the liver (*Figure 6F*), likely representing the tissue-resident Kupffer cell population, as well as a similar frequency of splenic macrophages (*Figure 6H*). However, upon infection, the livers of *Ccr2$^{-/-}$* mice had markedly less macrophages and drastically more neutrophils compared to the livers of WT mice (*Figure 6F, G*). This

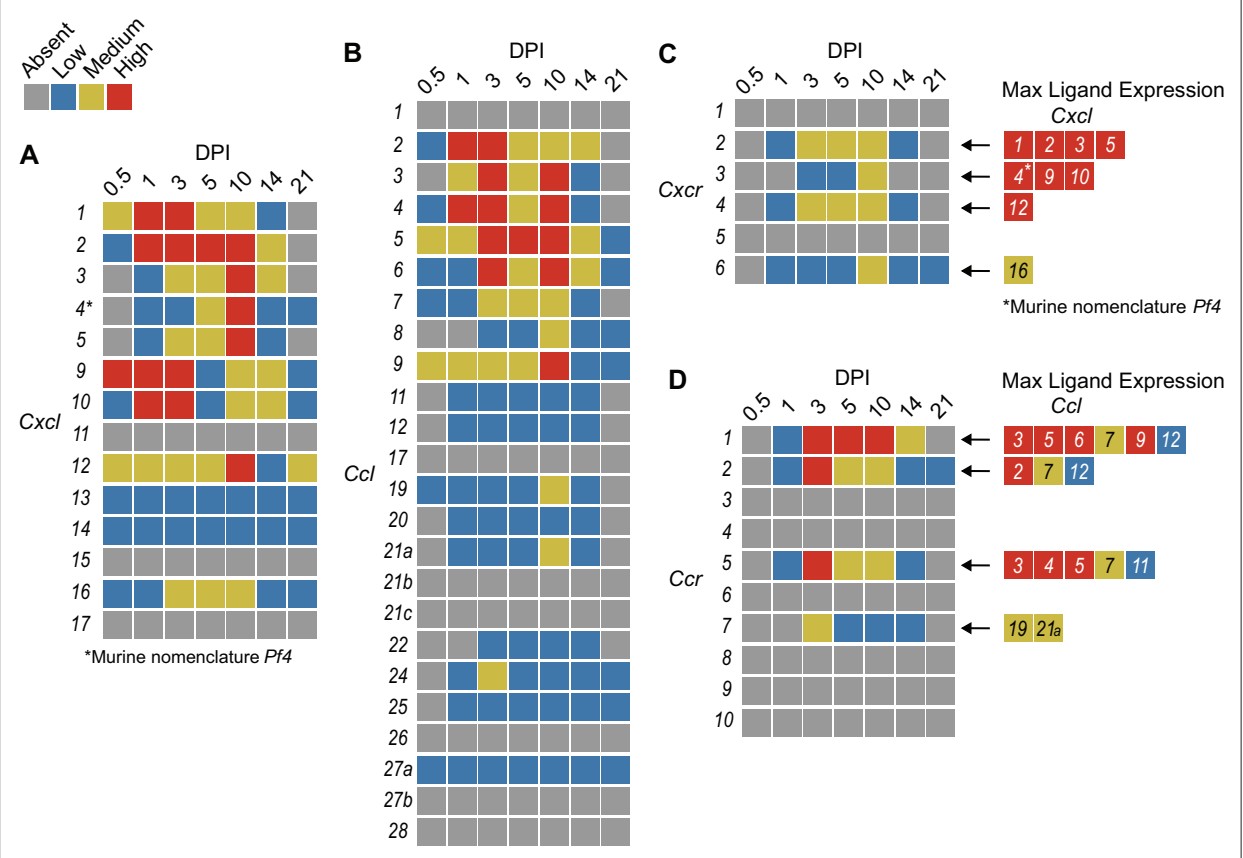

**Figure 4.** Qualitative heatmaps of chemokine and receptor expression during infection. Normalized expression in SpatialFeaturePlots was visually ranked as absent (gray), low (blue), medium (yellow), or high (red) for (**A**) CXCL family chemokines, (**B**) CCL family chemokines, (**C**) CXC chemokine receptors, and (**D**) CC chemokine receptors. Visual rankings were based on both the intensity of expression and the relative number of spots that expressed the gene. (**A, B**) Scale set at 0–3.0 expression; (**C–D**) Scale set at 0–2.0 expression. Arrows indicate ligand–receptor interactions. Ligands are color-coded based on the maximum expression level reached at any time during the course of infection.

The online version of this article includes the following figure supplement(s) for figure 4:

**Figure supplement 1.** Spatial expression of *Cxcl12*, *Cxcl13*, *Cxcl14*, and *Cxcl16*.

**Figure supplement 2.** Spatial expression of *Ccl3*, *Ccl4*, *Ccl5*, *Ccl6*, and *Ccl8*.

**Figure supplement 3.** Spatial expression of *Ccl9*, *Ccl11*, *Ccl19*, *Ccl20*, and *Ccl21a*.

**Figure supplement 4.** Spatial expression of *Ccl22*, *Ccl24*, *Ccl25*, and *Ccl27a*.

**Figure supplement 5.** Spatial expression of *Cxcr* family members.

**Figure supplement 6.** Spatial expression of *Ccr* family members.

trend was also observed in the spleen (*Figure 6H, I*) and blood (*Figure 6J, K*), showing that failure to recruit monocytes leads to enhanced neutrophil recruitment. Interestingly, infected *Ccr2*[−/−] mice did have slightly more macrophages in the liver, spleen, and blood compared to uninfected *Ccr2*[−/−] mice (*Figure 6F, H, J*), suggesting that loss of CCR2 does not completely abrogate monocyte recruitment. Alternatively, this expansion could represent emergency hematopoiesis and proliferation of pre-existing cell populations in these tissues (*Boettcher and Manz, 2017*).

## *C. violaceum* in the liver cannot be contained without macrophages

In our previous characterization of granulomas in WT mice, we identified three distinct zones using immunohistochemistry (IHC): necrotic core (NC), coagulative necrosis (CN), and macrophage zone (M) (*Harvest et al., 2023*). By 5 DPI, all three layers are distinctly visible through H&E staining (*Figure 7A*, *Figure 7—figure supplement 1*). Furthermore, we consistently see containment of *C. violaceum* within the necrotic core (*Figure 7B*), which overlaps with pronounced Ly6G staining (*Figure 7C*). Importantly,

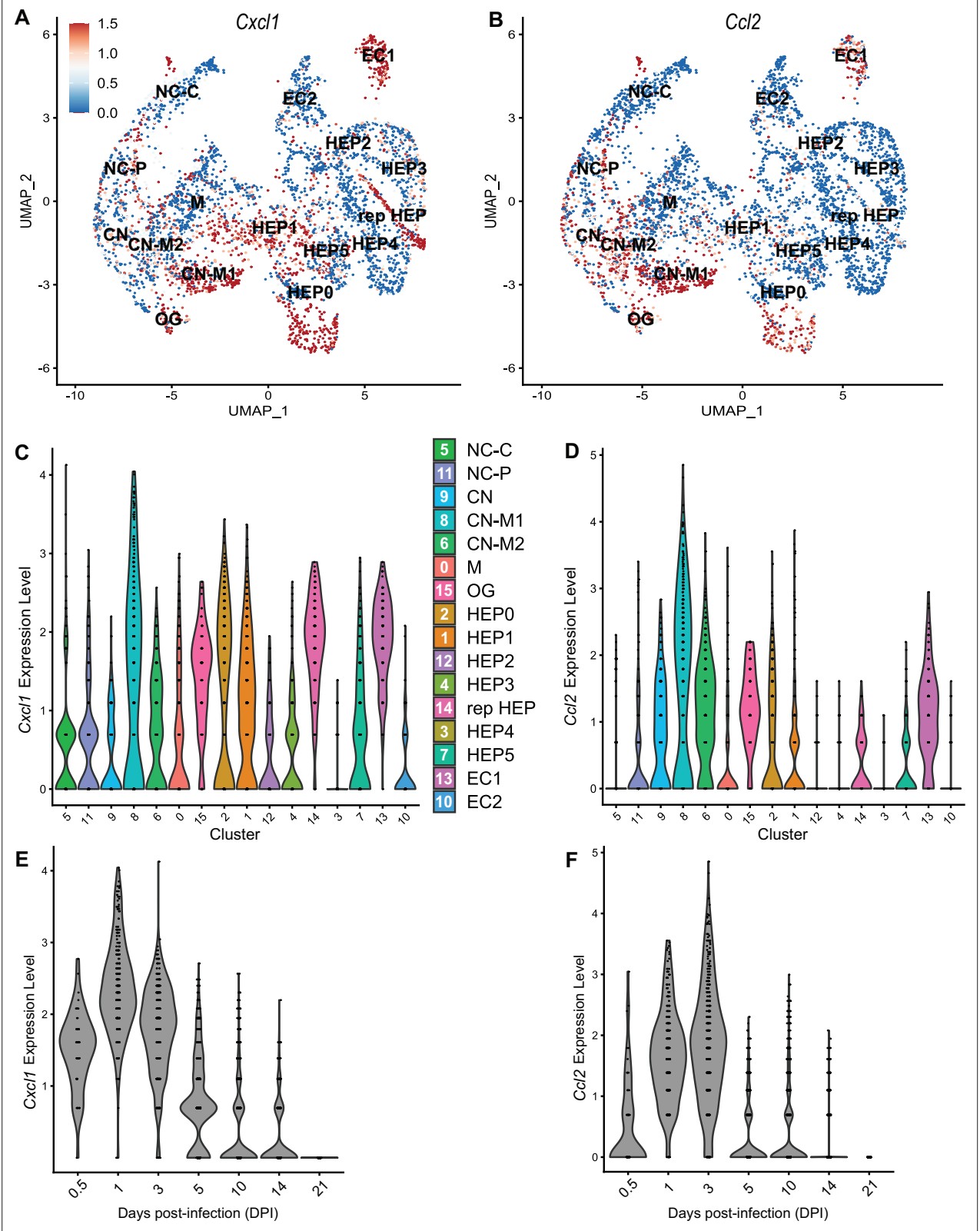

**Figure 5.** Chemokines involved in monocyte recruitment peak after chemokines involved in neutrophil recruitment. Comparative analysis of *Cxcl1* (**A, C**, and **E**) and *Ccl2* (**B, D**, and **F**). (**A, B**) UMAP plots of 16 unique clusters showing normalized expression level of each gene. Maximum expression level set to 1.5; annotated with cluster identity; macrophage zone (M), hepatocyte (HEP), representative HEP (rep HEP), necrotic core center (NC-C), NC-periphery (NC-P), coagulative necrosis (CN), CN-macrophage (CN-M), endothelial cell (EC), outside granuloma (OG). (**C, D**) Violin plots of 16

*Figure 5 continued on next page*

*Figure 5 continued*

unique clusters showing normalized expression level of each gene across all timepoints. (**E, F**) Violin plots of various days post-infection (DPI) showing normalized expression level of each gene within all clusters.

The online version of this article includes the following source data and figure supplement(s) for figure 5:

**Figure supplement 1.** Reparixin does not inhibit neutrophil chemotaxis into the liver of infected mice.

**Figure supplement 1—source data 1.** Bacterial burden data for *Figure 5—figure supplement 1B, D*.

**Figure supplement 1—source data 2.** Flow cytometry data for *Figure 5—figure supplement 1F–I*.

by 5 DPI the macrophage zone is clearly visible in WT mice, showing that macrophages surround the granuloma and form a protective zone between the coagulative necrosis zone and healthy hepatocytes outside the infected lesion (*Figure 7D*). Compared to WT mice, lesions in *Ccr2⁻/⁻* mice lack these distinct zones. Though *Ccr2⁻/⁻* mice had larger areas of necrotic debris, the coagulative necrosis zone was largely absent from most lesions (*Figure 7E*). In previous studies, we also observed sporadic clotting in WT mice (*Harvest et al., 2023*), and this clotting was even more abundant in *Ccr2⁻/⁻* mice (*Figure 7—figure supplement 1*). Excessive clotting, in addition to elevated bacterial burdens and sepsis, could also cause mortality in these mice by pulmonary embolism. Strikingly, lesions in *Ccr2⁻/⁻* mice had abnormal budding morphology, which stained very strongly for *C. violaceum* (*Figure 7F*) and Ly6G neutrophils (*Figure 7G*). In fact, many puncta that appear to be individual bacteria were visualized (*Figure 7—figure supplement 1*).

Though we were able to visualize the Kupffer cell population scattered throughout the liver, an organized macrophage zone was absent from the majority of lesions in *Ccr2⁻/⁻* mice (*Figure 7H*). These Kupffer cells are likely the CD68⁺ cells identified by flow cytometry (*Figure 6F*). A rare *Ccr2⁻/⁻* mouse that survived to 7 DPI also had few macrophages (*Figure 7—figure supplement 1*), in contrast to WT mice that display mature granulomas with thick macrophage zones at this timepoint (*Harvest et al., 2023*). Importantly, without distinct coagulative necrosis or macrophage zones, *C. violaceum* staining extends well outside the center of each lesion. In fact, numerous bacteria were identified in immune cells immediately adjacent to the healthy hepatocyte layer (*Figure 7F*). Importantly, we also observed these key differences through immunofluorescence, including larger necrotic cores with increased Ly6G staining, loss of organized macrophage zones, and bacterial staining directly adjacent to healthy hepatocytes (*Figure 7—figure supplement 3*). Furthermore, immunofluorescent staining of CCL2 revealed diffuse quantities in both WT and *Ccr2⁻/⁻* mice, with *Ccr2⁻/⁻* mice producing higher amounts of CCL2 in the liver and serum compared to WT mice at 3 DPI (*Figure 7—figure supplement 4*). This indicates that, especially in *Ccr2⁻/⁻* mice, the immune system is continuously calling for monocyte mobilization in response to *C. violaceum* infection. Taken together, the tissue staining, along with the elevated CFU burdens, suggests that monocyte recruitment fails without CCR2, and the lack of a macrophage zone leads to loss of bacterial containment. Despite the excessive number of neutrophils in the liver, spleen, and blood of *Ccr2⁻/⁻* mice (*Figure 6G, I, K*), these mice are unable to clear the infection and ultimately succumb.

Previously, we observed abnormal lesion architecture in *Casp1::Casp11* DKO and *Gsdmd⁻/⁻* mice with budding morphology and loss of bacterial containment (*Harvest et al., 2023*) that is remarkably similar to the architecture observed in *Ccr2⁻/⁻* mice (*Figure 7F*). However, the *Ccr2⁻/⁻* mice survive a few days longer and thus develop even larger lesions over time. Together, these data suggest that macrophage recruitment and pyroptosis are both essential in defense against, and containment of, *C. violaceum*. In addition, because the *Ccr2⁻/⁻* mice succumb in a timeframe similar to that seen with *Nos2⁻/⁻* mice, this supports our hypothesis that it is nitric oxide derived from granuloma macrophages that is specifically required for bacterial clearance. Altogether, these data indicate that without monocytes trafficking to the site of infection, *C. violaceum* is able to replicate and spread into adjacent hepatocytes, resulting in ever-expanding lesions. These in vivo data support the transcriptomics dataset and provide proof-of-concept that upregulated genes, specifically chemokines, are critical to the formation of the granuloma.

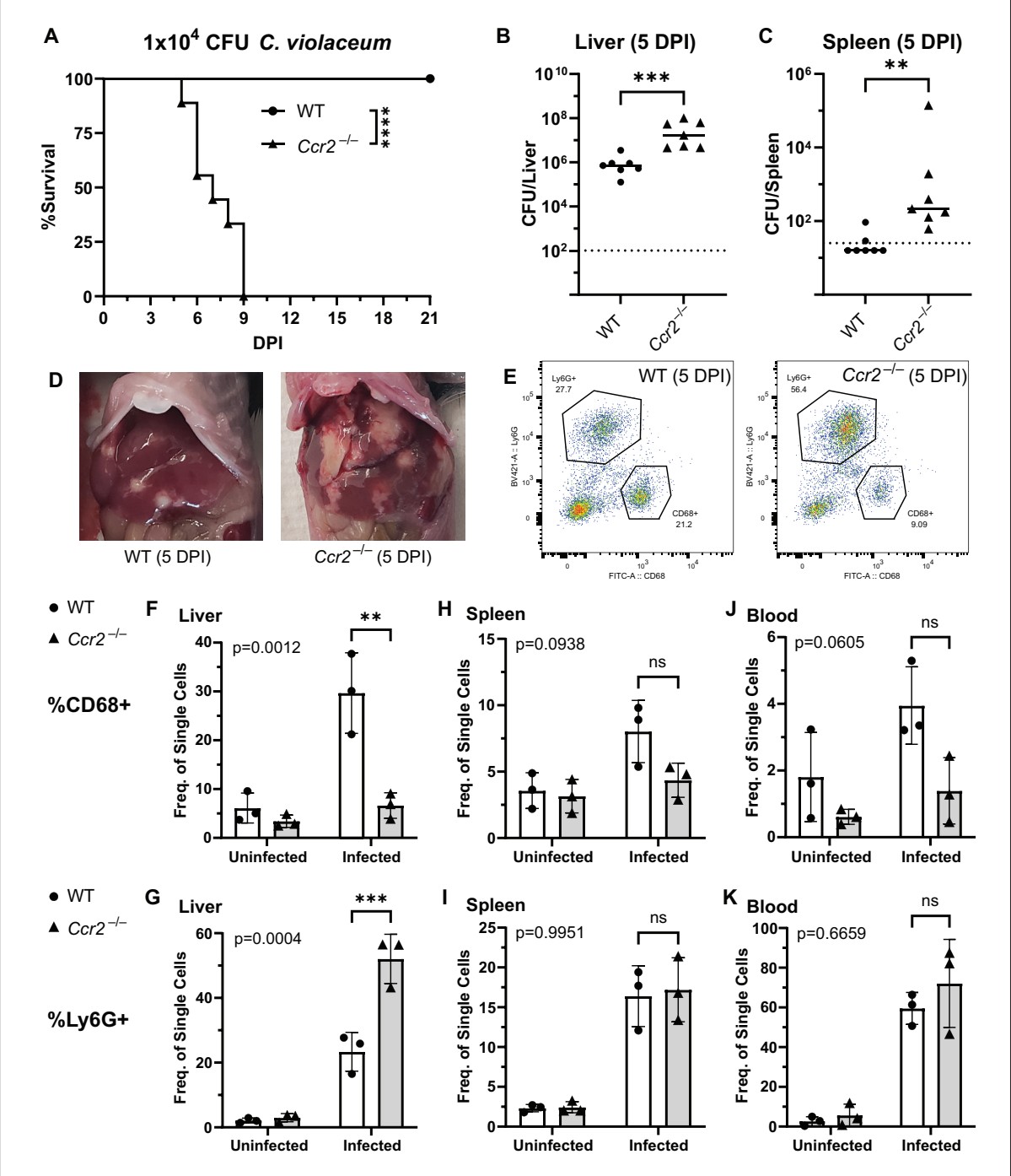

**Figure 6.** CCR2 and monocyte recruitment are essential for a successful granuloma response to *C. violaceum*. Wildtype (WT) and *Ccr2⁻/⁻* mice were infected intraperitoneally (IP) with 1 × 10⁴ CFU *C. violaceum*. (**A**) Survival analysis of WT (*N* = 10) and *Ccr2⁻/⁻* (*N* = 9) mice. Two experiments combined. Mantel–Cox test, ****p < 0.0001. (**B–K**) Livers and spleens were harvested 5 days post-infection (DPI). Bacterial burdens in the (**B**) liver and (**C**) spleen of WT and *Ccr2⁻/⁻* mice. Two experiments combined. Each dot represents one mouse. (**B**) Two-tailed *t* test (normally distributed data); ***p = 0.0002. (**C**) Mann–Whitney (abnormally distributed data); **p = 0.0012. Dotted line, limit of detection. Solid line, median. (**D**) Gross images of WT and *Ccr2⁻/⁻* livers 5 DPI. (**E**) Gating strategy for analysis of neutrophil (Ly6G⁺) and macrophage (CD68⁺) numbers via flow cytometry. Liver samples from infected mice shown. Frequency of CD68⁺ macrophages from single-cell gate in the (**F**) liver, (**H**) spleen, and (**J**) blood. Frequency of Ly6G⁺ neutrophils from single-cell gate in the (**G**) liver, (**I**) spleen, and (**K**) blood. (**F–K**) Three experiments combined using only female mice. Each dot represents one mouse, with 10,000 events collected per sample. Two-way ANOVA (for multiple comparisons to assess genotype and infection); key comparisons and p-values shown. Line represents mean ± standard deviation.

The online version of this article includes the following source data and figure supplement(s) for figure 6:

*Figure 6 continued on next page*

*Figure 6 continued*

**Source data 1.** Survival curve data for *Figure 6A*.

**Source data 2.** Bacterial burden data for *Figure 6B*.

**Source data 3.** Bacterial burden data for *Figure 6C*.

**Source data 4.** Flow cytometry data for *Figure 6F–K*.

**Figure supplement 1.** CCR2 and monocyte recruitment are essential for a successful granuloma response to *C. violaceum*.

**Figure supplement 1—source data 1.** Flow cytometry data for *Figure 6—figure supplement 1A–E*.

## Discussion

Here, we demonstrate that macrophages are essential for clearance of *C. violaceum* from the infected liver, and for protection against dissemination into the spleen. Loss of CCR2-dependent monocyte trafficking results in a loss of bacterial containment, ultimately leading to uncontrolled bacterial replication in the liver, evidenced by elevated CFU burdens and increased lesion size.

There are many questions that still remain about the individual and coordinated efforts of neutrophils and macrophages during infection with *C. violaceum*. It is likely that the tissue-resident Kupffer cells and infected hepatocytes are the first cells to sound the alarm, calling for neutrophils. The initial recruitment of neutrophils likely involves chemokines (i.e. CXCL1 and CXCL2) redundantly with other chemoattractants such as formylated peptides and leukotrienes. However, these neutrophils are unable to clear the infection despite being recruited in large numbers.

Based on our data, CCR2 is an essential chemokine receptor for monocyte trafficking in response to *C. violaceum*, but we have not yet determined which ligand(s) mediate this response. CCL2 and CCL7 can both bind to CCR2 to induce monocyte trafficking. Importantly, pro-inflammatory cytokines and PAMPs can induce CCL2 expression by most cell types (*Shi and Pamer, 2011*). In agreement, we see upregulation of *Ccl2* in several clusters and deposition of CCL2 protein in wide areas around granulomas, further suggesting that CCL2 may be a critical chemokine that promotes monocyte recruitment in response to *C. violaceum*. In contrast, *Ccl7* is expressed by fewer clusters, and to a lesser degree, and its expression is slightly delayed compared to *Ccl2*. Deletion of either ligand partially diminished monocyte trafficking in response to *Listeria monocytogenes* infection, but the individual role of each ligand was unclear (*Jia et al., 2008*). Future studies using *C. violaceum* could further elucidate the unique or redundant roles of CCL2 and CCL7. Lastly, adoptive transfer experiments in the context of *Listeria* infection showed that *Ccr2*$^{-/-}$ monocytes are still able to traffic to the site of infection in the spleen (*Serbina and Pamer, 2006*) and liver (*Shi et al., 2010*). During *C. violaceum* infection, we have not yet determined whether CCR2 is required for migration once monocytes have left the bone marrow, as CCR2 is required for this initial egress. We saw a subtle increase in the number of macrophages in the liver of infected *Ccr2*$^{-/-}$ mice. Though macrophage numbers in *Ccr2*$^{-/-}$ tissues remain considerably lower than seen in WT mice, there are two explanations for the subtle increase: (1) loss of CCR2 may not completely abrogate monocyte recruitment, as monocytes could be migrating via other chemokine receptors, or (2) tissue-resident macrophages, or even tissue-resident hematopoietic stem cells, could undergo emergency hematopoiesis and proliferate in response to infection (*Boettcher and Manz, 2017*). More studies are needed to assess the origin of this small population of macrophages in *Ccr2*$^{-/-}$ mice. Regardless, this small population of macrophages is not sufficient to protect against infection with *C. violaceum*.

In other granuloma models, the role of CCR2 is less clear. Loss of CCR2-dependent monocyte trafficking enhances clearance of *Y. pseudotuberculosis* (*Zhang et al., 2018*), which is a surprising result as typically macrophages would be expected to be important to clear infections. The role of CCR2 during *M. tuberculosis* infection is strain-dependent, and also varies depending on the dose and route of infection (*Dunlap et al., 2018*; *Peters et al., 2001*; *Scott and Flynn, 2002*). Though there are similarities between these infection models and *C. violaceum*, there are numerous differences. For example, expression of specific chemokines in response to *M. tuberculosis* differs from those we observe in response to *C. violaceum*, especially chemokines that attract T cells (*Kang et al., 2011*). A key concept in the *M. tuberculosis* field is that a delicate balance exists between cellular recruitment to control infection, and excess inflammation that causes disease symptoms (*Monin and Khader, 2014*). Furthermore, excess recruitment of monocytes to *M. tuberculosis*-induced granulomas leads to increased bacterial replication due to the ability of *M. tuberculosis* to inhibit

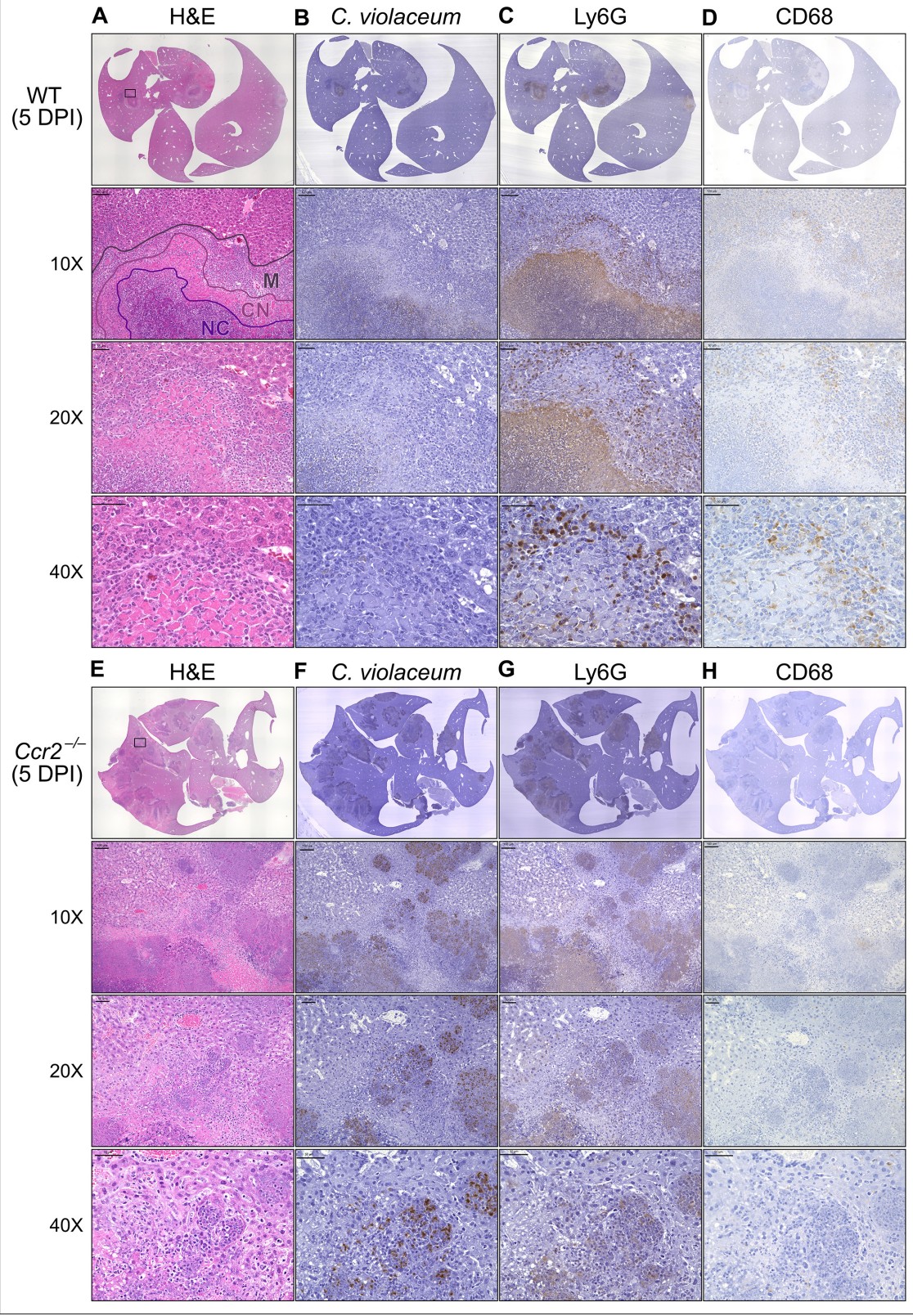

**Figure 7.** Loss of CCR2-dependent monocyte trafficking results in abnormal granuloma architecture and failure of bacterial containment. Wildtype (WT) and *Ccr2*⁻/⁻ mice were infected intraperitoneally (IP) with 1 × 10⁴ CFU *C. violaceum* and livers were harvested 5 days post-infection (DPI). Serial sections of livers stained by hematoxylin and eosin (H&E) or various immunohistochemistry (IHC) markers for (**A–D**) WT female and (**E–H**) *Ccr2*⁻/⁻ male. Necrotic

*Figure 7 continued on next page*

*Figure 7 continued*

core (NC), coagulative necrosis zone (NC), macrophage zone (M). For 10×, scale bar is 100 μm. For 20× and 40×, scale bar is 50 μm. Representative of two experiments with 2–4 mice per group, and multiple granulomas per section.

The online version of this article includes the following source data and figure supplement(s) for figure 7:

**Figure supplement 1.** Loss of CCR2-dependent monocyte trafficking results in abnormal granuloma architecture and failure of bacterial containment.

**Figure supplement 2.** *Ccr2⁻/⁻* mice have increased necrosis and clotting.

**Figure supplement 3.** Loss of CCR2-dependent monocyte trafficking results in abnormal granuloma architecture and failure of bacterial containment.

**Figure supplement 4.** *Ccr2⁻/⁻* mice have increased CCL2 in the liver and serum.

**Figure supplement 4—source data 1.** ELISA data for *Figure 7—figure supplement 4D, E*.

degradation within phagosomes in which it resides (*Domingo-Gonzalez et al., 2016*; *Slight and Khader, 2013*). In contrast, *C. violaceum* appears to lack sufficient virulence factors to enable it to replicate within macrophages (*Batista and da Silva Neto, 2017*). Importantly, while *M. tuberculosis* bacterial burdens plateau at 21 DPI, almost all mice clear *C. violaceum* by this timepoint. Though decades of research have been dedicated to investigating *M. tuberculosis*, fewer studies involving other granuloma-inducing pathogens have been performed. As we continue to study the cellular mechanisms that allow for successful granuloma formation and clearance of *C. violaceum*, it will be interesting to compare the two pathogens, as future studies could shed light on key differences that result in successful pathogen clearance.

In WT mice, neutrophil recruitment wanes as the granuloma matures, which coincides with clearance of *C. violaceum*. However, in the *Ccr2⁻/⁻* mice, we see elevated neutrophil numbers at 5 DPI, suggesting that neutrophils are continuously recruited in the absence of macrophages. Under normal circumstances, endocytosis of chemokines by endothelial cells helps to diminish chemokine gradients, limiting prolonged neutrophil recruitment (*Kolaczkowska and Kubes, 2013*). Future studies could investigate the various signals that diminish neutrophil recruitment in WT mice during clearance, and why this fails in *Ccr2⁻/⁻* mice. Another interesting component of the granuloma response is the spatial arrangement of neutrophils and macrophages within the granuloma. In vitro studies found that CCR1 and CCR5 differentially affected monocyte localization within a transwell system, implying that a system exists for fine-tuning the exact location of macrophages within inflamed tissues (*Shi and Pamer, 2011*). These receptors are highly upregulated in the *C. violaceum*-induced granuloma and are thus good candidates for balancing the localization of macrophages between the coagulative necrosis zone and healthy tissue outside the granuloma.

Lastly, this dataset inspires a number of new hypotheses related to granuloma resolution and tissue repair after bacterial clearance. Chemokines undergo a variety of post-translational modifications, such as glycosylation, nitration, citrullination, and proteolytic cleavage, which can either enhance or abrogate their activity (*Vanheule et al., 2018*). For example, nitration of CCL2 and CCL3 by peroxynitrite was shown to reduce monocyte and neutrophil chemotaxis, respectively (*Sato et al., 1999*; *Sato et al., 2000*). Furthermore, binding to atypical receptors can also affect chemokine availability, representing another mechanism to resolve inflammation (*Hansell et al., 2006*; *Ulvmar et al., 2011*). Of particular interest is the implication of matrix metalloproteinases (MMPs) in regulating chemokine functions. MMPs can not only directly cleave chemokines, they can also cleave various chemokine-binding proteins that help establish the chemokine gradient (*Parks et al., 2004*). Several studies have found that MMPs can cleave chemokines to alter their function, either increasing or decreasing their receptor binding activity. For example, MMP-2 cleaves both CXCL12 and CCL7, abolishing their ability to induce chemotaxis (*McQuibban et al., 2000*; *McQuibban et al., 2001*); importantly, all three of these genes are upregulated during *C. violaceum* infection (*Table 1*, *Table 3*, *Table 4*). Furthermore, MMP-2 and MMP-9 have been extensively studied in the context of lung inflammation, both of which are important to limit tissue damage (*Greenlee et al., 2006*). MMP-9 has also been shown to promote or inhibit liver fibrosis and wound repair, depending on the context (*Feng et al., 2018*). An unsolved mystery during infection with *C. violaceum* is how the chemotaxis of neutrophils and monocytes is abrogated when the infection is cleared, and how wound repair and resolution is initiated. Future studies could characterize the role of MMPs during resolution, especially MMP-9 and its various targets in relation to wound repair.

Analysis of a spatial transcriptomics dataset revealed the upregulation of many chemokines and their receptors during murine infection with *C. violaceum*. Here, we show that CCR2 is an essential chemokine receptor for monocyte trafficking, which enables the formation of mature granulomas with organized macrophage zones. Importantly, loss of organized macrophages leads to loss of bacterial containment. This work has given new insight into the function of chemokines during granuloma formation, and this model of *C. violaceum*-induced granuloma formation will be useful in exploring the unique and redundant roles of chemokines during infection.

# Materials and methods

**Key resources table**

| Reagent type (species) or resource | Designation | Source or reference | Identifiers | Additional information |
|---|---|---|---|---|
| Strain, strain background (*Mus musculus*) | Wildtype C57BL/6 mice (WT) | Jackson Laboratory (West Grove, PA) | Ref# 000664 | |
| Strain, strain background (*Mus musculus*) | *Ccr2*^RFP (*Ccr2*^−/−) | Jackson Laboratory | Ref# 017586 | |
| Strain, strain background (Bacteria) | *Chromobacterium violaceum (C. violaceum)* | ATCC (Manassas, VA) | Ref# 12472 | |
| Antibody | Rat anti-mouse Ly6G monoclonal (IA8) in BV421 | BD Biosciences (Franklin Lakes, NJ) | Ref# 562737 | 1:300 (FC) |
| Antibody | Rat anti-mouse monoclonal (FA-11) CD68 in FITC | BioLegend (San Diego, CA) | Ref# 137005 | 1:300 (FC) |
| Antibody | Rabbit anti-*C. violaceum* polyclonal | Cocalico Biologicals (Denver, PA) | Custom polyclonal antibody | 1:2000 (IHC, IF) |
| Antibody | Rat anti-mouse Ly6G monoclonal (IA8) | BioLegend | Ref# 127601 | 1:300 (IHC) |
| Antibody | Rabbit anti-mouse CD68 polyclonal | Abcam (Waltham, MA) | Ref# ab125212 | 1:200 (IHC) |
| Antibody | Rat anti-mouse CD68 monoclonal (FA-11) in Alexa Fluor 488 | Abcam | Ref# ab201844 | 1:100 (IF) |
| Antibody | Rat anti-mouse Ly6G monoclonal (IA8) in Alexa Fluor 647 | BioLegend | Ref# 127610 | 1:100 (IF) |
| Antibody | Rabbit anti-mouse MCP1 (CCL2) polyclonal | Abcam | Ref# ab315478 | 1:100 (IF) |
| Antibody | Goat anti-rabbit secondary polyclonal in Alexa Fluor 594 | Invitrogen (Waltham, MA) | Ref# A32740 | 1:1000 (IF) |
| Commercial assay or kit | Avidin/Biotin Blocking Kit | Vector Laboratories (Newark, CA) | Ref# SP-2001 | |
| Commercial assay or kit | SignalStain Boost IHC Detection Reagent (HRP, Anti-Rabbit) | Cell Signaling (Danvers, MA) | Ref# 8114 | |
| Commercial assay or kit | ImmPRESS HRP Goat Anti-Rat Detection Kit | Vector Laboratories | Ref# MP-7404 | |
| Commercial assay or kit | DAB Substrate Kit, HRP | Vector Laboratories | Ref# SK-4100 | |
| Commercial assay or kit | H&E Stain Kit (Modified Mayer's Hematoxylin and Bluing Reagent) | Abcam | Ref# ab245880 | |
| Commercial assay or kit | MCP-1/CCL2 Mouse Uncoated ELISA Kit | Thermo Scientific (Waltham, MA) | Ref# 88-7391-22 | |
| Chemical compound, drug | Reparixin | MedChemExpress (Monmouth Junction, NJ) | Ref# HY-15251 | |
| Software, algorithm | RStudio | Posit PBC (Boston, MA) | | |
| Software, algorithm | FlowJo | BD Biosciences | | |

*Continued on next page*

*Continued*

| Reagent type (species) or resource | Designation | Source or reference | Identifiers | Additional information |
|---|---|---|---|---|
| Software, algorithm | Prism 9 | GraphPad (Boston, MA) | | |
| Software, algorithm | Fiji | ImageJ (Burleson, TX) | | |
| Other | Collagenase Type IV | Gibco | Ref# 17104019 | Tissue dissociation media |
| Other | 1× DMEM, +4.5 g/l D-Glucose, +L-Glutamine, +110 mg/l Sodium Pyruvate | Gibco | Ref# 11995-065 | Cell culture media |
| Other | 1× RPMI Medium 1640, +L-Glutamine | Gibco | Ref# 11875-093 | Cell culture media |
| Other | PenStrep +10,000 units/ml Penicillin, +10,000 µg/ml Streptomycin | Gibco | Ref# 15140-122 | Antibiotics |
| Other | HyClone Characterized Fetal Bovine Serum | Cytiva (Marlborough, MA) | Ref# SH30396.03 | Cell culture media |
| Other | 1× DPBS, -Calcium Chloride, -Magnesium Chloride | Gibco | Ref# 14190-144 | Cell culture media |
| Other | 70 µm Cell Strainers | Genesee Scientific (El Cajon, CA) | Ref# 25-376 | Tissue dissociation reagent |
| Other | 40 µm Cell Strainers | Genesee Scientific | Ref# 25-375 | Tissue dissociation reagent |
| Other | Percoll | GE Healthcare (Chicago, IL) | Ref# 17-0891-01 | Tissue dissociation reagent |
| Other | 1× RBC Lysis Buffer | eBioscience | Ref# 00-4333-57 | Flow cytometry reagent |
| Other | Falcon Round-Bottom Polystyrene Test Tubes | Thermo Scientific | Ref# 14-959-1A | Flow cytometry tubes |
| Other | Mouse BD Fc Block | BD Biosciences | Ref# 553142 | Blocking reagent; used at 1 µg (FC), 2% (IF) |
| Other | Intracellular Fixation & Permeabilization Buffer | eBioscience | Ref# 88-8824-00 | Flow cytometry reagent |
| Other | 10% Neutral Buffered Formalin | VWR (Radnor, PA) | Ref# 16004–128 | Histology reagent |
| Other | 16% Paraformaldehyde | VWR | Ref# 15710S | Immunofluorescence reagent |
| Other | Sucrose | Sigma-Aldrich | Ref# S1888 | Immunofluorescence reagent |
| Other | Tissue-Tek O.C.T. Compound | Sakura (Torrance, CA) | Ref# 4583 | Immunofluorescence reagent |
| Other | Epredia Xylene | Fisher Chemical | Ref# 99-905-01 | Immunohistochemistry reagent |
| Other | ImmEdge Pen | Vector Laboratories | Ref# H-4000 | Immunohistochemistry reagent |
| Other | Normal Goat Serum Blocking Solution, 2.5% | Vector Laboratories | Ref# S-1012 | Immunohistochemistry reagent |
| Other | SignalStain Antibody Diluent | Cell Signaling | Ref# 8112 | Immunohistochemistry reagent |
| Other | Permount | Fisher Chemical | Ref# SP15-100 | Immunohistochemistry reagent |
| Other | T-PER Tissue Protein Extraction Reagent | Thermo Scientific | Ref# 78510 | Tissue dissociation reagent |
| Other | Sulfuric Acid | Ricca Chemical (Arlington, TX) | Ref# 8310-32 | ELISA Stop Buffer |
| Other | Fluoroshield with DAPI | Sigma-Aldrich | Ref# F6057 | Immunofluorescence reagent |

## Analysis of spatial transcriptomics dataset

Tissues from infected mice were harvested at the indicated timepoints, which were chosen based on key events observed via H&E staining (*Harvest et al., 2023*). Spatial data were generated in *Harvest et al., 2023* using the 10X Genomics Visium Platform. We were most interested in the immune cells present within the distinct zones of each lesion, and the adjacent healthy hepatocytes. Therefore, we used Loupe Browser v7.0 to visualize the H&E-stained tissues and manually select spots

of interest. We deselected spots that were distant from infected lesions, while selecting the lesions and surrounding healthy hepatocytes. To account for cell-to-cell variation, especially across tissues, pre-processing included normalization using sctransform (*Hafemeister and Satija, 2019*). To further analyze the spatial transcriptomics dataset of the selected spots, we used the Seurat package in RStudio to analyze gene expression over time and space. UMAP plots, SpatialDimPlots, SpatialFeaturePlots, ggplots, and Violin plots were all used to visualize normalized gene expression data.

## Ethics statement and mouse studies

All mice were housed in groups of two to five according to IACUC guidelines at Duke University (under protocols A018-23-01 and A043-20-02). WT C57BL/6 mice (referred to as WT; from Jackson Laboratories) or $Ccr2^{RFP}$ mice (referred to as $Ccr2^{-/-}$; originally generated in *Saederup et al., 2010*) were used as indicated. Mice were moved to a BSL2 facility a minimum of 3 days prior to treatment. For experiments involving infection, mice were monitored every 24 hr for signs of illness. After the appearance of symptoms, mice were monitored every 12 hr. Mice showing sever signs of illness were euthanized according to previously established euthanasia criteria.

## Treatment of mice with reparixin

Stock solutions of reparixin were prepared in PBS with gentle warming for a final concentration of 20 mg/kg in 200 µl PBS. Mice were injected subcutaneously with 200 µl of appropriate reparixin stock or with 200 µl PBS (control).

## Preparation of inoculum

Bacteria were grown overnight on brain heart infusion (BHI) agar plates (*C. violaceum* ATCC strain 12472) at 37°C and stored at room temperature for no more than 2 weeks. To prepare infectious inocula, bacteria were cultured in 3 ml BHI broth with aeration overnight at 37°C before being diluted in PBS to indicated infectious inoculum.

## In vivo infections

For in vivo infections, 8- to 10-week-old, age- and sex-matched mice were infected as previously described (*Harvest et al., 2023*). Mice were infected intraperitoneally with indicated number of bacteria in 200 µl PBS. Whole livers and spleens were harvested at indicated timepoints.

## Plating for CFUs

At the indicated DPI, mice were euthanized and the spleen and liver were harvested for quantification of bacterial burdens as previously described (*Harvest et al., 2023*). Briefly, spleens were placed in a 2-ml homogenizer tube with 1 large metal bead and 1 ml sterile PBS, and whole livers were placed in a 7-ml homogenizer tube with 1 large metal bead and 3 ml sterile PBS. Tube weights were recorded before and after tissue harvest to normalize CFUs/volume/tissue. After homogenizing, 1:5 serial dilutions were performed in sterile PBS, and dilutions were plated on BHI in triplicate or quadruplicate. The following day, bacterial colonies were counted and CFU burdens calculated.

## Flow cytometry

At the indicated DPI, mice were euthanized and the spleen, liver, and whole blood were harvested for flow cytometry as previously described (*Harvest et al., 2023*). For experiments involving whole blood, cardiac puncture was used to collect 100 µl whole blood prior to perfusion with PBS through the vena cava as described in *Mendoza et al., 2022*. Briefly, whole livers were minced on ice using scissors and incubated in digestion buffer (100 U/ml Collagenase Type IV in DMEM supplemented with 1 mM $CaCl_2$ and 1 mM $MgCl_2$) for 40 min in a 37°C water bath with intermittent vortexing. Digested livers were homogenized through a 40-µm cell strainer, followed by washing with RPMI (supplemented with 1× Pen/Strep and 1% FBS) and centrifugation at 300 × $g$ for 8 min. Leukocytes from the liver were further isolated using a Percoll gradient where samples were resuspended in 45% Percoll with an 80% Percoll underlay, and spun at 800 × $g$ for 20 min with no brake. For spleens, tissues were mechanically homogenized through a 70-µm strainer, followed by washing and centrifugation at 300 × $g$ for 5 min. Red blood cells were lysed with 1× RBC Lysis Buffer according to product manual (note: whole blood was stained with Ly6G at room temperature prior to RBC lysis. Blood samples were treated identically

to liver and spleen samples thereafter). Liver and spleen samples were counted using trypan blue, and $1 \times 10^6$ cells per tissue per mouse were stained for various cell markers: Mouse BD Fc Block (1 μg), rat anti-mouse Ly6G in BV421 (1:300), rat anti-mouse CD68 in FITC (1:300) for 30 min. For CD68, staining was performed using Intracellular Fixation & Permeabilization Buffer according to product manual. For each sample, 10,000 events were acquired on a BD LSRFortessa X-20 Cell Analyzer at the Duke Flow Cytometry Core Facility. Samples were analyzed using FlowJo (for Windows, version 10.7.1).

## ELISA

At 3 DPI, mice were euthanized and whole blood (about 500 μl) and a piece of liver were harvested for ELISA. Whole blood and liver tissue were collected as described for flow cytometry, except whole blood was allowed to coagulate at room temperature for 30 min before separating the serum through centrifugation at 10,000 × $g$ for 5 min at 4°C. Serum was collected and stored at −80°C until analysis. Following perfusion, a piece of liver tissue containing visible granulomas was harvested and stored at −80°C. Liver pieces were then homogenized as described for CFU enumeration, except 30 μl T-PER per 5 mg tissue was used in place of PBS. Homogenates were incubated on ice for 2 hr prior to analysis. Serum and liver samples were analyzed for CCL2 according to ELISA kit protocol, and plates read on a BioTek Synergy H1 microplate reader. Calculations were performed in Excel.

## Histology and IHC

To prepare paraffin-embedded tissues, whole livers were harvested at the indicated DPI and submerged in 20 ml of 10% neutral buffered formalin. Samples were gently inverted every day for a minimum of 3 days before being transferred to tissue cassettes and given to the Histology Research Core at the University of North Carolina at Chapel Hill. The research core performed tissue embedding, serial sectioning, slide mounting, and staining of H&E samples. For IHC, serial sections were then processed and stained as described in *Harvest et al., 2023*. Washes were performed in 1× TBS-T. Primary antibodies were diluted in SignalStain antibody diluent, and included: rabbit anti-*C. violaceum* (1:2000), rat anti-Ly6G (1:300), and rat anti-CD68 (1:200). Slides were incubated in primary antibody overnight at 4°C in a humidity chamber. Prior to staining with secondary antibody, endogenous peroxidase activity was blocked using 3% $H_2O_2$. Slides were incubated in secondary antibody (SignalStain Boost HRP anti-rabbit or ImmPRESS HRP anti-rat) at room temperature for 30 min. Incubation with DAB Substrate Kit was performed for 30 s to 2 min, depending on the intensity of signal. Slides were counter-stained with hematoxylin for 5 s to 1 min, depending on the intensity of the DAB, and then dipped in bluing reagent for 1 min. After dehydration, slides were covered with Permount mounting medium and a coverslip. Importantly, WT slides and *Ccr2*−/− slides were stained side-by-side.

## Immunofluorescence

To prepare frozen tissues, livers were perfused with 2% paraformaldehyde (PFA) (diluted in PBS) through the vena cava (*Mendoza et al., 2022*). Individual lobes of the liver were excised and stored in 2% PFA overnight at 4°C. Tissues were subsequently stored for 48 hr in 30% sucrose at 4°C. Finally, tissues were frozen in O.C.T. compound on dry ice before being stored at −80°C. Slides with 5-μm thick tissue sections were prepared using a Thermo Scientific CryoStar NC70 Cryostat, and slides were stained as described in *Harvest et al., 2023*.

## Microscopy

Histology, IHC, and immunofluorescence samples were analyzed on a KEYENCE All-in-One Microscope BZ-X800. For immunofluorescence imaging, exposure times were set so that uninfected liver appeared negative, and exposure times were maintained between samples. Immunofluorescent images were further analyzed in BZ-X800 Analyzer. Histology and IHC images were further analyzed in Fiji by ImageJ.

## Statistics

Statistical analysis was performed using GraphPad Prism 9.5.1. For survival analysis, the Mantel–Cox test was used to compare WT and *Ccr2*−/− mice. For bacterial burdens, data were first assessed for normality using the Shapiro–Wilk test. For two groups, a two-tailed $t$-test (or Mann–Whitney for abnormally distributed data) was used. For more than two groups, a two-way ANOVA was used.

## Acknowledgements

We thank Michael Dee Gunn for key discussions regarding chemokines and Ashley Moseman for sharing mice. We thank Heather N Larson for mouse colony maintenance. Original graphics created with BioRender.com. This research was funded by NIH grants AI139304, AI136920, AI175078 (EAM), and AI133236-04S1 (CKH). The APC was funded by startup funds from Duke University School of Medicine.

## Additional information

### Funding

| Funder | Grant reference number | Author |
|---|---|---|
| National Institute of Allergy and Infectious Diseases | AI139304 | Edward A Miao |
| National Institute of Allergy and Infectious Diseases | AI136920 | Edward A Miao |
| National Institute of Allergy and Infectious Diseases | AI175078 | Edward A Miao |
| National Institute of Allergy and Infectious Diseases | AI133236-04S1 | Carissa K Harvest |

The funders had no role in study design, data collection, and interpretation, or the decision to submit the work for publication.

### Author contributions

Megan E Amason, Data curation, Formal analysis, Software, Conceptualization, Validation, Investigation, Visualization, Methodology, Writing – original draft, Writing – review and editing; Cole J Beatty, Formal analysis, Software, Conceptualization; Carissa K Harvest, Formal analysis, Funding acquisition, Methodology; Daniel R Saban, Supervision, Project administration; Edward A Miao, Data curation, Supervision, Funding acquisition, Writing – review and editing

### Author ORCIDs

Megan E Amason ⓘ https://orcid.org/0009-0000-2834-5834
Edward A Miao ⓘ https://orcid.org/0000-0001-7295-3490

### Ethics

All mice were housed in groups of two to five according to IACUC guidelines at Duke University (under protocols A018-23-01 and A043-20-02). For experiments involving infection, mice were monitored every 24 hr for signs of illness. After the appearance of symptoms, mice were monitored every 12 hr. Mice showing sever signs of illness were euthanized according to previously established euthanasia criteria.

Reviewer #1 (Public review): https://doi.org/10.7554/eLife.96425.3.sa1
Reviewer #2 (Public review): https://doi.org/10.7554/eLife.96425.3.sa2
Author response https://doi.org/10.7554/eLife.96425.3.sa3

## Additional files

### Supplementary files

- MDAR checklist
- Source code 1. Streamlined code for analysis using RStudio.

## Data availability

All study data are included in the article as source data files, and the spatial transcriptomics dataset is deposited with Dryad (https://doi.org/10.5061/dryad.zpc866thz). Streamlined RStudio code is included in *Source code 1*.

The following dataset was generated:

| Author(s) | Year | Dataset title | Dataset URL | Database and Identifier |
|---|---|---|---|---|
| Amason M, Miao E, Harvest C | 2024 | Spatial transcriptomics of an innate granuloma in a mouse infection model with *Chromobacterium violaceum* | https://doi.org/10.5061/dryad.zpc866thz | Dryad Digital Repository, 10.5061/dryad.zpc866thz |

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
