## [Editor Report · eLife assessment]

This **valuable** study advances the understanding of granuloma formation by identifying a key chemokine receptors in containing infection by a specific species of bacteria. The evidence supporting this is **solid**, providing a spatial transcriptomic dataset spanning granuloma formation and resolution by a specific species of bacteria. The work should be of interest to microbiologists and immunologists.

---

## [Referee Report · Reviewer #1 (Public review)]

Amason et al. investigated the formation of granulomas in response to Chromobacterium violaceum infection, aiming to uncover the cellular mechanisms governing the granuloma response. They identify spatiotemporal gene expression of chemokines and receptors associated with the formation and clearance of granulomas, with a specific focus on those involved in immune trafficking, generating a valuable spatial transcriptomic reference. By analyzing the presence or absence of chemokine/receptor RNA expression, they infer the importance of immune cells in resolving infection. Despite observing increased expression of neutrophil-recruiting chemokines, treatment with reparixin (an inhibitor of CXCR1 and CXCR2) did not inhibit neutrophil recruitment during infection. Focusing on monocyte trafficking, they found that CCR2 knockout mice infected with *C. violaceum* were unable to form granulomas, ultimately succumbing to infection.

Readers should note that due to the resolution of the spatial data, it is difficult to associate gene expression differences with individual cell types; the authors focus instead on changes in chemokines and chemokine receptors, and perform experiments to evaluate the importance of CCR2.

Comments on the revised version:

The authors have addressed all of my previous comments.

---

## [Referee Report · Reviewer #2 (Public review)]

Summary:

In this study Amason et al employ spatial transcriptomics and intervention studies to probe the spatial and temporal dynamics of chemokines and their receptors, and their influence on cellular dynamics in *C. violaceum* granulomas. As a result of their spatial transcriptomic analysis, the authors narrow in on the contribution of neutrophil-and monocyte-recruiting pathways to host response. This results in the observation that monocyte recruitment is critical for granuloma formation and infection control, while neutrophil recruitment via CXCR2 may be dispensable.

Strengths:

Since *C. violaceum* is a self-limiting granulomatous infection, it makes an excellent case study for 'successful' granulomatous inflammation. This stands in contrast to chronic, unproductive granulomas that can occur during *M. tuberculosis* infection, sarcoidosis, and other granulomatous conditions, infectious or otherwise. Given the short duration of *C. violaceum* infection, this study specifically highlights the importance of innate immune responses in granulomas.

Another strength of this study is the temporal analysis. This proves to be important when considering the spatial distribution and timing of cellular recruitment. For example, the authors observe that the intensity and distribution of neutrophil and monocyte recruiting chemokines vary substantially across infection time and correlate well with their previous study of cellular dynamics in *C. violaceum* granulomas.

The intervention studies done in the last part of the paper bolster the relevance of the authors' focus on chemokines. The authors provide important negative data demonstrating the null effect of CXCR1/2 inhibition on neutrophil recruitment during *C. violaceum* infection. That said, the authors' difficulty with solubilizing reparixin in PBS is an important technical consideration given the negative result. On the other hand, monocyte recruitment via CCR2 proves to be indispensable for granuloma formation and infection control.

Weaknesses:

There are several shortcomings that limit the impact of this study. The first is that the cohort size is very limited. While the transcriptomic data is rich, the authors analyze just one tissue from one animal per timepoint. This assumes that the selected individual will have a representative lesion and prevents any analysis of inter-individual variability. Granulomas in other infectious diseases, such as schistosomiasis and tuberculosis, are very heterogeneous. The authors do assert that in *C. violaceum* infection granulomas are very consistent in their composition and kinetics, alleviating, in part, this concern.

Another caveat to these data is the limited or incompletely informative data analysis. This dataset has been previously published with more extensive and broad characterization. Here, the authors use Visium in a more targeted manner to interrogate certain chemokines and cytokines. While this is a great biological avenue, key findings rely on qualitative inspection of gene expression overlaid on to images or data that has been qualitatively binned or thresholded. Upon revision the authors did supplement their analyses with important information, such as the top expressed genes in each Visium cluster and the dynamic range of RNA counts retrieved across clusters.

Furthermore, the authors are underutilizing the spatial information provided by Visium with no spatial analysis conducted to quantify the patterning of expression patterns or spatial correlation between factors. The authors acknowledge the challenge of conducting this analysis given the variable size and geometry of the granulomas. In future studies, this can be overcome through size- or distance-based normalization or spatial clustering approaches that evaluate local neighborhood composition across different scales.

Impact:

The author's analysis helps highlight the chemokine profiles of protective, yet host protective granulomas. As that authors comment on in their discussion, these findings have important similarities and differences with other notable granulomatous conditions, such as tuberculosis. Beyond the relevance to *C. violaceum* infection, these data can help inform studies of other types of granulomas and hone candidate strategies for host-directed therapy strategies.

---

## [Author Response]

The following is the authors’ response to the original reviews.

**Reviewer #1 (Public Review):**
Amason et al. investigated the formation of granulomas in response to Chromobacterium violaceum infection, aiming to uncover the cellular mechanisms governing the granuloma response. They identify spatiotemporal gene expression of chemokines and receptors associated with the formation and clearance of granulomas, with a specific focus on those involved in immune trafficking. By analyzing the presence or absence of chemokine/receptor RNA expression, they infer the importance of immune cells in resolving infection. Despite observing increased expression of neutrophil-recruiting chemokines, treatment with reparixin (an inhibitor of CXCR1 and CXCR2) did not inhibit neutrophil recruitment during infection. Focusing on monocyte trafficking, they found that CCR2 knockout mice infected with *C. violaceum* were unable to form granulomas, ultimately succumbing to infection.The spatial transcriptomics data presented in the figures could be considered a valuable resource if shared, with the potential for improved and clarified analyses. The primary conclusion of the paper, that *C. violaceum* infection in the liver cannot be contained without macrophages, would benefit from clarification.

We thank the reviewer for their time and effort in evaluating our manuscript.

While the spatial transcriptomic data generated in the figures are interesting and valuable, they could benefit from additional information. The manual selection of regions of granulomas for analysis could use additional context - was the rest of the liver not sequenced, or excluded for other reasons? Including a healthy liver in the analysis could serve as a control for any lasting effects at the final time point of 21 days.

We revised the text in the methods section to include additional information about manual selection of regions. The entire tissue section was sequenced, but using H&E as a guide, we manually selected each representative lesion and a surrounding layer of healthy hepatocytes at each timepoint. We agree that an uninfected control could be useful, however we did not include an uninfected mouse in the experiment because we were most interested in the cells that make up the granuloma, not hepatocytes outside the lesion. Additionally, we find that in the 21 DPI timepoint the surrounding hepatocytes appear to have returned to a homeostatic transcriptional state; at 21 DPI the majority of mice have undetectable CFU burdens.

Providing more context for the scalebars throughout the spatial analyses, such as whether the data are raw counts or normalized based on the number of reads per spatial spot, would be helpful for interpretation, as changes in expression could signal changes in the numbers of cells or changes in the gene expression of cells.

The scalebars for the SpatialFeaturePlots display the normalized gene expression values. The data are normalized based on the number of reads per spatial spot, using the sctransform method published in (Hafemeister & Satija, 2019). We agree that the changes in expression could result from changes in cell numbers and/or changes in gene expression on a per cell basis. However, the sctransform method is designed to preserve biological variation while minimizing technical effects observed in transcriptomics platforms. Regardless of the heterogeneity of sequencing depth, it is clear from these plots that gene expression changes dynamically over time and space, which was the focus of our analysis. We have updated the figure legends to clarify scalebar units, and revised the methods section.

In Figure 4, qualitative measurements are valuable, but having an idea of the raw data for a few of the pursued chemokines/receptors would aid interpretation

All of the SpatialFeaturePlots utilized to generate Figure 4 have been included in the manuscript, either in the main figures or in the supplemental figures. For example, the SpatialFeaturePlots of *Cxcl4*, *Cxcl9*, and *Cxcl10* are all in Figure 4 – figure supplement 1.

In Figure 4 it would also be beneficial to clarify whether the reported values are across all clusters and consider focusing on clusters with the greatest change in expression.

Figure 4 summarizes the expression of each gene at each timepoint for the entire selected area, independently of cluster identity. Different clusters do show variability in the relative change in expression. To better show these data, we have included an additional graphic that summarizes the top twenty upregulated genes for each cluster, many of which include chemokines (new Table 4). The average log2FC values for each of these genes can be found in Table 4 – source data 1.

Figures 5E and F would benefit from clarification regarding the x-axis units and whether the expression levels are summed across all clusters for each time point

Figures 5E and 5F display the normalized gene expression values for all spots (independent of cluster identity) at each timepoint. We have updated the figure legend to reflect this clarification.

Additionally, information on the sequencing depth of the samples would be helpful, particularly as shallow sequencing of RNA can result in poor capture of low-expression transcripts.

We agree with the reviewer that sequencing depth is an additional factor to take into consideration. We have included an additional supplemental figure (Figure 1 – figure supplement 1A-B) to display raw counts spatially at the various timepoints, and within each cluster.

Regarding the conclusion of the essentiality of macrophages in granuloma formation, it may be prudent to further investigate the role of macrophages versus CCR2. Consideration of experiments deleting macrophages directly, instead of CCR2, could provide more definitive evidence of the necessity of macrophage migration in containing infections.

While CCR2 is expressed on a number of other cells besides monocytes, it is well-documented that loss of CCR2 results in accumulation of monocytes in the bone marrow and a significant reduction in the blood-monocyte population. As a result, monocytes are not recruited to the site of infection in numerous prior publications in the field; we confirm this as shown by flow cytometry and IHC. Nonetheless, future studies will aim to rescue *Ccr2*–/– mice via adoptive transfer of monocytes to further show that monocyte-derived macrophages are essential for defense against infection. We also intend to perform clodronate depletion experiments at various timepoints, however, clodronate will also deplete Kupffer cells and has off-target effects on neutrophils. Overall, the established importance of CCR2 for monocyte egress from the bone marrow and our observation that the macrophage ring fails to form give us sufficient confidence to conclude that monocyte-derived macrophages are essential for this innate granuloma.

Analyzing total cell counts in the liver after infection could provide insight into whether the decrease in the fraction of macrophages is due to decreased numbers or infiltration of other cell types...

Our flow data suggest that the decrease in macrophages in *Ccr2*–/– mice is due to both a decrease in macrophage number and an increase in the infiltration of other cell types (namely neutrophils). To better illustrate this, we now include an additional quantification of the total cell counts in the liver and spleen (new Figure 6 – figure supplement 1), which supports our conclusion that *Ccr2*–/– mice have a defect in granuloma macrophage numbers. We have also repeated the experiment to reach sufficient numbers to perform statistical analysis (revised Figure 6F–K).

**Reviewer #2 (Public Review):**
Summary:In this study, Amason et al employ spatial transcriptomics and intervention studies to probe the spatial and temporal dynamics of chemokines and their receptors and their influence on cellular dynamics in *C. violaceum* granulomas. As a result of their spatial transcriptomic analysis, the authors narrow in on the contribution of neutrophil- and monocyte-recruiting pathways to host response. This results in the observation that monocyte recruitment is critical for granuloma formation and infection control, while neutrophil recruitment via CXCR2 may be dispensable.

We thank the reviewer for their thoughtful comments and suggestions.

Strengths:Since *C. violaceum* is a self-limiting granulomatous infection, it makes an excellent case study for 'successful' granulomatous inflammation. This stands in contrast to chronic, unproductive granulomas that can occur during *M. tuberculosis* infection, sarcoidosis, and other granulomatous conditions, infectious or otherwise. Given the short duration of *C. violaceum* infection, this study specifically highlights the importance of innate immune responses in granulomas.Another strength of this study is the temporal analysis. This proves to be important when considering the spatial distribution and timing of cellular recruitment. For example, the authors observe that the intensity and distribution of neutrophil- and monocyte-recruiting chemokines vary substantially across infection time and correlate well with their previous study of cellular dynamics in *C. violaceum* granulomas.The intervention studies done in the last part of the paper bolster the relevance of the authors' focus on chemokines. The authors provide important negative data demonstrating the null effect of CXCR1/2 inhibition on neutrophil recruitment during *C. violaceum* infection. That said, the authors' difficulty with solubilizing reparixin in PBS is an important technical consideration given the negative result...

We agree with the reviewer, and the limited solubility of reparixin and other chemokine-receptor inhibitors is a major caveat of this study and others in the field. In future studies, there are several other inhibitors that could be used to further assess the role of CXCR1/2.

On the other hand, monocyte recruitment via CCR2 proves to be indispensable for granuloma formation and infection control. I would hesitate to agree with the authors' interpretation that their data proves macrophages are serving as a physical barrier from the uninvolved liver. It is possible and likely that they are contributing to bacterial control through direct immunological activity and not simply as a structural barrier.

We agree that macrophages do not form a physical or structural barrier, a word that implies epithelial-like function. Instead, we agree that macrophages mostly act immunologically. We revised the text to remove the term barrier.

Weaknesses:There are several shortcomings that limit the impact of this study. The first is that the cohort size is very limited. While the transcriptomic data is rich, the authors analyze just one tissue from one animal per time point. This assumes that the selected individual will have a representative lesion and prevents any analysis of inter-individual variability.Granulomas in other infectious diseases, such as schistosomiasis and tuberculosis, are very heterogeneous, both between and within individuals. It will be difficult to assert how broadly generalizable the transcriptomic features are to other *C. violaceum* granulomas...

We thank the reviewers for highlighting this key difference between granulomas in other infectious diseases, and granulomas induced by *C. violaceum*. Based on many prior experiments, we observe that *C. violaceum*-induced granulomas are very reproducible between and within individuals (highlighted in our previous publication). As this is a major advantage of this model system, we chose specific timepoints based on key events that consistently occur in the majority of lesions assessed at each timepoint, allowing us to be confident in the selection of representative granulomas. However, it is worth noting that granulomas within an individual mouse are seeded and resolved somewhat asynchronously. This did indeed affect our spatial transcriptomic data, as the 7 DPI timepoint was not histologically representative of a typical 7 DPI granuloma. Therefore, we excluded the 7 DPI timepoint from our analyses.

Furthermore, this undermines any opportunity for statistical testing of features between time points, limiting the potential value of the temporal data.

We agree with the reviewer that there is much more characterization and quantification that can be done. As demonstrated by the abundance of spatial and temporal data for the chemokine family alone, the spatial transcriptomics dataset is rich and will likely supply us with many years of analyses and investigations. Our current approach is to use the spatial transcriptomics dataset as a hypothesis-generating tool, followed by in vivo studies that seek to uncover physiological relevance for our observations. In the current paper, the strength of the spatial transcriptomic data for CCL2, CCL7 and their receptor CCR2 prompted us to study *Ccr2*–/– mice. These mice then prove the relevance of the spatial transcriptomic data. In regard to conclusions about temporal changes in chemokine expression, in this manuscript we do not make conclusions that CCL2 is important at one timepoint but not another. We are characterizing the broad temporal trends of expression in order to cast a broad net to inform future in vivo studies. There is much work for us to do to explore all the induced chemokines and their receptors.

Another caveat to these data is the limited or incompletely informative data analysis. The authors use Visium in a more targeted manner to interrogate certain chemokines and cytokines. While this is a great biological avenue, it would be beneficial to see more general analyses considering Visum captures the entire transcriptome. Some important questions that are left unanswered from this study are:What major genes defined each spatial cluster?...

The initial characterization of each spatial cluster was performed in Harvest et al., 2023. In brief, we used a mixture of published single-cell sequencing data, histological-based parameters, and ImmGen to define each cluster. We have not re-stated those methods in the current manuscript, but instead reference our prior paper.

What were the top differentially expressed genes across time points of infection?...

Though the top differentially expressed genes for each cluster can be informative in some situations, we chose a more targeted approach because of the obvious importance of chemokines. Nonetheless, we have included an additional graphic that summarizes the top twenty upregulated genes for each cluster (new Table 4). The average log2FC values for each of these genes can be found in Table 4 – source data 1.

Did the authors choose to focus on chemokines/receptors purely from a hypothesis perspective or did chemokines represent a major signature in the transcriptomic differences across time points?

We chose to focus on chemokines because of their obvious importance for recruitment of immune cells. They were also among the highest induced genes in the spatial transcriptome (new Table 4).

In addition to the absence of deep characterization of the spatial transcriptomic data, the study lacks sufficient quantitative analysis to back up the authors' qualitative assessments...

See above comment regarding statistical comparisons.

Furthermore, the authors are underutilizing the spatial information provided by Visium with no spatial analysis conducted to quantify the patterning of expression patterns or spatial correlation between factors.

Several factors make quantification challenging. Lesions grow considerably in size in the first few days of infection, and then shrink in size in the latter days. This makes quantification challenging between timepoints. Radial quantification is also challenging due to the irregular shapes of each granuloma (see comment below for further discussion). Most importantly, the key next experiments are to validate the importance of each chemokine and receptor in vivo. Once we know which ones are the most important, this will justify putting more effort into spatial quantitative analysis and patterning of expression for those chemokines.

Impact:The author's analysis helps highlight the chemokine profiles of protective, yet host protective granulomas. As the authors comment on in their discussion, these findings have important similarities and differences with other notable granulomatous conditions, such as tuberculosis. Beyond the relevance to *C. violaceum* infection, these data can help inform studies of other types of granulomas and hone candidate strategies for host-directed therapy strategies.
**Recommendations for the authors:**

**Reviewer #2 (Recommendations For The Authors):**
The Visium analysis would be strengthened by(1) Showing several histology examples of granulomas at each timepoint to help aid the reader in seeing how 'representative' each Visium sample is...

These histological analyses are performed in our previous manuscript, and indeed were a crucial aspect of the initial characterization of the spatial transcriptomics dataset, which was performed in Harvest et al., 2023. Full liver sections are shown in that paper at each timepoint, and readers can see that the architecture is highly reproducible.

(2) Validating their results in other tissues, either with Visium or with more targeted assays for their study's key molecules, such as immunohistochemistry or in situ hybridization

We agree on the importance of validation studies and have plans to perform single-cell RNA sequencing experiments to further enhance resolution. With key genes in mind, we then plan to perform more in vivo studies to assess physiological relevance of upregulated genes in specific cell types.

At the very least it would be important to validate the expression of CXCL1 and CXCL2 in other tissues and at the protein level, given the importance of those findings

We think that the reviewer is asking us to validate that CXCL1 and CXCL2 are actually expressed given the negative reparixin data. However, if we do prove that they are expressed, this will not resolve whether they have critical roles in neutrophil recruitment. To prove this, we would need either a better CXCR2 inhibitor or *Cxcr2* knockout mice. Therefore, we are saving further exploration for the future. Regarding validating other chemokines, we establish that CCR2 is critical, and we now show by immunofluorescence and ELISA (new Figure 7 – figure supplement 4) that CCL2 is highly expressed in WT mice, and *Ccr2*–/– mice actually have strongly elevated CCL2 expression at 3 DPI compared to WT mice.

In Figure 1B, the UMAP here is largely uninformative. To display the clusters, the authors should instead show a heatmap or equivalent visualization of which genes defined each cluster. It would be helpful for the authors to also write out the full name of each cluster before using the abbreviations shown.

Please see our previous comment about the initial characterization of clusters performed in Harvest et al., 2023, which details the characteristic genes for each cluster. We have written the full names of each cluster in the legend of Figure 1.

In Figure 1C the authors, use a binary representation of whether a cluster is present or not at a particular time point. However, the spot size is arbitrary, and the colors of the dots are the same as the cluster color code. It is not clear what threshold the authors (or SpatialDimPlots) use to declare a given cluster is present at a given time point. Therefore, this chart does not give any sense of the extent of each cluster's presence at each time. The authors should revisualize these data to display the abundance of each cluster at each timepoint. This could simply be done by adjusting the size of the circle or using a more traditional heatmap.

We have now updated this graphic to display the extent of a cluster’s presence, with the size of each dot corresponding to the abundance of each cluster.

In Figures 2 and 3 the authors describe the kinetics of each chemokine by cluster. While the dynamic expression is evident in the images, it is challenging to determine which clusters are driving expression in the absence of cluster annotation in those figures. The authors should support their visual findings with quantification of each factor in each cluster across time points.

In Figure 5, violin plots are shown for *Cxcl1* and *Ccl2* that depict gene expression by each cluster. However, because each capture area is approximately 50 µm in diameter, the data do not achieve single-cell resolution and are not as informative as one would hope. Therefore, violin plots for each chemokine were not shown, though we have generated these graphics. We did not add these graphics to the revision because we did not think readers would generally want to see several pages of violin plots in the supplement. As mentioned, we plan to do single-cell RNA sequencing to further assess chemokine expression by each cell type present within the granulomas at key timepoints.

With respect to the lack of spatial analysis, the authors describe certain transcript signals (ie. peripheral region versus central region of the granuloma) across each lesion. To back up these qualitative assertions, the authors could use line profiles from the center of each granuloma to the outside to plot the variation in expression of each transcript over radial space. This would provide a more direct way to determine the spatial coordination between various transcripts.

We considered using line profiles to quantify spatial variation within each lesion at each timepoint. However, this was exceptionally challenging due to the asymmetrical nature of some lesions, and the size discrepancy at different timepoints as the granulomas grow (during infection) and shrink (during resolution). When attempting to decide where to draw the line profiles, we determined that this approach did not enhance our analyses beyond using the cluster overlay and H&E to identify and interrogate different clusters.

The data visualization in Figure 4 seems unnecessarily confusing. The authors put the transcriptomic signal into categories of 'absent', 'low', 'medium', and 'high.' Why not simply use a continuous scale? The data would also benefit from hierarchical clustering of the heatmap rows to highlight chemokines and their receptors with similar expression patterns across time.

We considered using a continuous scale as suggested by the reviewer. However, we chose not to create a continuous scale because quantitation is challenging due to the size changes in the lesions over time, such that larger lesions have greater inclusion of surrounding hepatocytes as well as necrotic cores, which would dilute the signal if averaged with the active immunologic granuloma zones. Figure 4 was intended to simplify the entirety of the SpatialFeaturePlots in an easy-to-digest manner, to aid in hypothesis generation as we consider the potential function of each chemokine and receptor in this model. We chose to organize each chemokine ligand based on family, maintaining a numerical order to allow Figure 4 to serve as a quick reference for anyone who is interested in a particular chemokine ligand or receptor.

Do the authors feel confident in the transcriptomic signal coming from regions of necrosis? Given that many of their bright signals are coming from within clusters annotated as necrosis or necrosis-adjacent this raises an important technical consideration. Can the authors use the H&E image to estimate the cellular density (based on nuclear counts) in each region annotated by Visium? Are there any studies supporting the accurate performance of spatial transcriptomic methods in necrosis? Necrosis can be a source of non-specific binding during in situ hybridization assays.

The reviewer raises a good point. A defining characteristic of the areas of necrosis is the lack of defined cell borders, with faded or absent nuclei. In these regions, it is impossible to estimate cellular density. Given these concerns, we have included an additional figure (new Figure 1 – figure supplement 1A-B) to display raw counts in each cluster across all timepoints. Though regions of necrosis do display lower read quantity compared to other areas, we are still confident in the positive transcriptomic signal coming from adjacent regions because there are plenty of negative examples in which expression is not detected. In other words, temporal and spatial upregulation of key genes is still observed in the tissues, and future experiments will aim to interrogate the physiological relevance of each gene, while validating the spatial transcriptomics data with other methodologies.

The methods should include a much more detailed description of the tissue preparation and collection for the Visium experiment. The section on the computational analysis of the Visium data is also extremely limited. At a minimum, the authors should include details on how they performed clustering of the Visium regions.

The detailed description of tissue preparation, computational analysis, and clustering is in our previous manuscript, from which this dataset originates. We can add a direct quote of the methodology if the reviewer requests.

The cluster labels in Figure 5 A-B are very difficult to see. Furthermore, it would help if the authors displayed the annotated cluster names (ie. Those shown in 5C) instead of their numerical coding for a more direct interpretation of the data.

We agree and have updated this figure with annotated cluster names.

The scale bars in Figure 7 are very difficult to see.

The scale bars in histology images were kept small intentionally so as not to occlude data, and eLife is an online-only, digital media platform which allows readers to sufficiently zoom on high-resolution histology images. We have increased the DPI resolution for histology images to further aid in visualization.

The information presented in Tables 2 and 3 is greatly appreciated and will really help guide the reader through the analyses.

We assembled this information for our own learning about chemokines and hope that it is useful for the reader.